# Features of Chromosome Introgression from *Gossypium barbadense* L. into *G. hirsutum* L. during the Development of Alien Substitution Lines

**DOI:** 10.3390/plants11040542

**Published:** 2022-02-18

**Authors:** Marina Feliksovna Sanamyan, Shukhrat Umarovich Bobohujayev, Sharoffidin Sayfiddinovich Abdukarimov, Abdusalom Khasanboyevich Makamov, Olga Gennadevna Silkova

**Affiliations:** 1Department of Biology, National University of Uzbekistan named M. Ulugbek, University St. 4, Tashkent 100174, Uzbekistan; bobohujayev@mail.ru; 2Center of Genomics and Bioinformatics UAS, University St. 2, Kibray Region, Tashkent 111215, Uzbekistan; sharofiddinabdukarimov@gmail.com (S.S.A.); amakamov@gmail.com (A.K.M.); 3Institute of Cytology and Genetics, SB RAS, Lavrentiev Av. 10, 630090 Novosibirsk, Russia; silkova@bionet.nsc.ru

**Keywords:** cotton, *G. barbadense*, *G. hirsutum*, monosomic lines, chromosome substitution lines, molecular markers, translocation

## Abstract

The creation of *G. barbadense* L./*G. hirsutum* L. chromosome-substitution lines is an important method to transfer agronomically valuable traits from *G. barbadense* into *G. hirsutum*. In this study, 30 monosomic lines of *G. hirsutum* from the Cytogenetic Collection of Uzbekistan, created in the genotypic background of line L-458, were used in crosses with the *G. barbadense* line Pima 3-79 to create substitution lines. In the course of this work, new monosomic lines were identified for chromosome **12** and monotelodisome **6** of the A_t_subgenome and for chromosomes **17**, **21,** and **22** of the D_t_subgenome using chromosome-specific SSR markers and a well-defined tester set of cotton translocation lines (USA). Compared to those in the F_1_ hybrids, a strong decrease in the crossing and setting rates was found in the BC_1_F_1_ backcross lines, with the substitution of chromosomes **2**, **4**, **6**, **7**, and **12** of the A_t_subgenome and **17**, **18**, **21**, and **22** of the D_t_subgenome. The F_1_ and BC_1_F_1_ offspring from interspecific crosses differed in their transmission of univalents. Despite the regular pairing of chromosomes and the high meiotic index, interspecific aneuploid hybrids were characterized by a decrease in pollen fertility, which may indicate hidden structural variability in these genomes that did not affect meiotic division. The identification of chromosomes using chromosome-specific SSR markers in the early stages of plant development has greatly accelerated the detection of monosomic plants. The analysis of morphobiological traits revealed that monosomic F_1_ hybrids were more similar to the donor line, while BC_1_F_1_ hybrids were more similar to the recurrent parent but also showed previously undetected traits.

## 1. Introduction

Cotton belongs to the genus *Gossypium,* which comprises approximately 53 species [1,2]. The genus *Gossypium* is unusually diverse; 46 species have been assigned to eight cytologically and geographically distinct groups with diploid genomes (A, B, C, D, E, F, G, and K) of 2n = 26, and 7 species have been assigned to a group with a tetraploid genome (AADD) of 2n = 52 [1,2,3].

In diploid species, significant chromosome rearrangements have occurred during evolution [4,5]. Allopolyploid species likely arose 1–2 million years ago as a result of the spread of a taxon containing the A genome in the New World, followed by hybridization of this taxon with a local diploid containing the D genome and doubling of the number of chromosomes [6].To date, the alleged donors of the genomes of tetraploid species are thought to come from African or Asian species with the A genome and from a species similar to the American diploid species with the D genome [1,7]. It has been shown that the subgenomes of allotetraploids—D_t_ and A_t_ (‘t’ indicates tetraploid)—are collinear with the genomes of *G. raimondii* Ulbr. and *G. arboreum* L., respectively [8,9].

The nascent A_t_D_t_ allopolyploid spread throughout the American tropics and subtropics, diverging into seven species: AD1 = *G. hirsutum* L., AD2 = *G. barbadense* L., AD3 = *G. tomentosum* Nutt. ex Seem., AD4 = *G. mustelinum* Miers ex Watt., AD5 = *G. darwinii* Watt., AD6 = *G. ekmanianum* Wittm., and AD7 = *G. stephensii* J. Gallagher, C. Grover, and Wendel. [1,10]. Two allopolyploid cotton species *G. hirsutum* L. or Upland cotton and *G. barbadense* L. (or Pima cotton) were independently domesticated [11]. Genome doubling in tetraploids has led to a variety of molecular genetic interactions, including different rates of genome evolution, intergenomic gene transfer, and changes in gene expression [6,12,13,14].

To date, an increase in cotton yields has also been achieved through traditional breeding and intervarietal crossing. Cultivars have been created with better yield potential, higher fibre quality, and the ability to withstand changing climatic conditions, as well as early ripening times, a tolerance to high fertilizer contents, and the ability to efficiently use soil moisture reserves [15]. However, most of these varieties were obtained by selection from a narrow gene pool and adapted to certain soil and climatic conditions. Thus, modern cultivated cotton shows a reduction in genetic diversity, which has resulted in a decrease in fibre quality and an increase in genetic vulnerability due to the close relationship of high-yielding varieties.

Enrichment of the *G. hirsutum* genome with alleles from other economically valuable cotton species is very important [15,16,17]. For example, *G. tomentosum* is characterized as having heat resistance, while *G. mustelinum* and *G. stocksii* Mast. are characterized as having pest and disease resistance [18]. The wild species *G. longicalyx* J.B. Hutch and B.J.S. Lee (F1 genome) could be used as a donor of desirable traits for fibre fineness, length, and strength [19].

It is known that *G. barbadense* is less productive and has a narrower range of agronomic adaptability, but it has significantly higher quality fibre (length, strength, and fibre fineness) than cultivated varieties of *G. hirsutum*, although the latter gives a higher yield. Given their complementary agronomic characteristics, numerous attempts have been made to hybridize the two species within the framework of traditional breeding. However, this work has been unsuccessful.

In cotton, interspecific hybrid progeny often have poor agronomic traits, distorted segregation, sterility, mote formation, and limited recombination due to genome incompatibility. A comparative analysis of the genomes of *G. barbadense* and *G. hirsutum* revealed large-scale inversions on various chromosomes of the A_t_ and D_t_subgenomes [5]. The genetic effects of inversions are manifested in the suppression of the recombination of the inverted regions themselves and the regions adjacent to them, reducing population diversity and leading to population divergence. Thus, inversions between *G. hirsutum* and *G. barbadense* restrict recombination in F_1_ hybrids, limiting genomic exchanges. Generally, F_1_ hybrids of *G. hirsutum* × *G. barbadense* are fertile, but the F_2_ phenotypes and subsequent generations are biased towards one of the parents due to the manifestation of genetic breakdown.

Another method of transferring agronomically valuable traits from *G. barbadense* to *G. hirsutum* has been the creation of lines with introgressed segments of individual chromosomes from *G. barbadense* and lines with chromosome substitutions of *G. barbadense* L./*G. hirsutum* L. [20,21,22,23].

D.M. Stelly et al. [22] constructed 17 chromosome- or chromosomal-arm-substitution lines of *G*. *barbadense* in the genetic background of the *G*. *hirsutum* line Texas Marker-1 (TM-1). Previous analyses of CS lines have associated many novel traits with the substitution of chromosomes and chromosome segments.

Thus, the study of these lines made it possible to determine that the substitution of homologous chromosomes from *G. hirsutum* L. into *G. barbadense* L. (CS-B02, CS-B04,CS-B16, CS-B17, CS-B22Lo, CS-B22sh, and CS-B25) had an effect on fibre elongation, fibre yield, fibre strength, and micronair, when compared with the original lines TM-1 and Pima 3-79 [24,25,26].

In creating *G. barbadense* L./*G. hirsutum* L.-substituted lines [22,23], monosomic individuals of TM-1 *G. hirsutum* L. were used or targeted transfer of certain chromosomes of *G. barbadense* L. In this work, a similar scheme was used, and monosomic lines of *G. hirsutum* from the Cytogenetic Collection of Uzbekistan were crossed with the *G. barbadense* line Pima 3-79 to create *G. hirsutum*/*G. barbadense* substitution lines with improved agronomic traits. Identification of univalent chromosomes was carried out at the early stages of cotton-plant development using chromosome-specific molecular markers.

Previously, we obtained new monosomic cotton lines in a single genotypic background of the highly inbred *G. hirsutum* L. line L-458 (A_t_D_t_ genome) [27,28,29,30]; these lines resulted in long-term inbreeding and selection with the cotton cultivar 108-F. Univalent chromosomes of these lines were identified using a well-defined tester set of cotton translocation lines (USA) [31,32]. The identification of most monosomic lines was confirmed using SSR markers at the Center for Genomics and Bioinformatics of the Academy of Sciences of the Republic of Uzbekistan [30,31]. As a result, four monosomic lines from the Cytogenetic Collection of Uzbekistan (Mo11, Mo16, Mo19, and Mo93) were found to have chromosome **2** of the A_t_subgenome; 16 monosomic lines (Mo7, Mo31, Mo38, Mo58, Mo59, Mo60, Mo69, Mo70, Mo71, Mo72, Mo73, Mo75, Mo76, Mo79, Mo81, and Mo89) had chromosome **4** of the A_t_subgenome; five monosomic lines (Mo13, Mo34, Mo67, Mo92, and Mo95) had chromosome **6** of the A_t_subgenome; Mo27 had chromosome **7** of the A_t_subgenome; Mo48 line had chromosome **18** of the D_t_subgenome; and monotelodisomic line Mo21 lacks an arm of chromosome **11** of the A_t_subgenome [31,32,33,34].

Fivefold backcrossing was used to create *G. hirsutum*/*G. barbadense* near-isogenic substitution lines [23]. Backcrossing was also used to prevent “genetic breakdown” in these interspecific crosses [23]. Using this technique, a set of chromosome-segment-introgression lines (CSILs) were obtained [20,21,23]. To our knowledge, data on the peculiarities of chromosome behaviour in interspecific aneuploid hybrids when obtaining these lines have not been published. Such information is necessary to understand the collinearity of *G. hirsutum* chromosomes with *G. barbadense* chromosomes, which reflects their compensatory ability, as well as to understand the contribution of individual chromosomes of these two species to the manifestation of “genetic breakdown” during interspecies crossing and subsequent backcrossing.

The aim of this work was to study interspecific aneuploid F_1_ hybrids obtained in a different genotypic background from crosses of monosomic cotton lines of the inbred *G. hirsutum* line L-458 with the *G. barbadense* L. line Pima 3-79, as well as their BC_1_F_1_ backcross offspring. In the course of this work, new monosomic lines of *G. hirsutum* L. were obtained; the identification of univalent chromosomes was carried out using molecular- genetics and translocation markers; and the crossing of monosomic lines with Pima 3-79 was performed. Additionally, the setting and germination of hybrid seeds were studied; the meiotic behaviour of chromosomes in monosomic F_1_ and BC_1_F_1_ plants has been investigated; and univalent chromosomes in F_1_ and BC_1_F_1_ hybrids have been identified using molecular genetic markers. Finally, a comparative morphobiological analysis of aneuploid F_1_ and BC_1_F_1_ hybrids was carried out and the prospect of using them to obtain lines with substitutions of specific chromosomes was assessed.

## 2. Results

### 2.1. Development Scheme for Obtaining Cotton Chromosome-Substitution Lines

We developed a scheme for creating cotton lines with chromosome substitutions. According to the scheme (Figure 1), the previously identified *G. hirsutum* L. aneuploid lines were crossed with the *G. barbadense* L. donor line Pima 3-79; the hybrid offspring from each crossing were studied at the metaphase I stage of meiosis; and SSR markers were used to confirm the chromosome identity. Furthermore, aneuploidy F_1_ hybrid plants were crossed with the aneuploid *G. hirsutum* L. line (recurrent parent), and the resulting BC_1_F_1_ backcross hybrids were studied using molecular marker analysis (SSR) at the plantlet stage to identify aneuploid BC_1_F_1_ hybrids, which were confirmed by cytogenetic analysis. The aneuploid BC_1_F_1_ hybrids were then recrossed with the recurrent parent. Furthermore, the obtained BC_2_F_1_ backcross hybrids were analysed using molecular markers (SSRs) at the plantlet stage. The identified BC_2_F_1_ aneuploid hybrids were re-examined at the metaphase I stage of meiosis to detect aneuploid BC_2_F_1_ hybrids. This work was repeated up to the fifth generation using backcrossing. Then, aneuploid hybrid BC_5_F_1_ plants were self-pollinated, and euploid plants from the self-pollinated BC_5_F_1_ generation were used as the founders of chromosome-substituted lines (Figure 1).

Double screening of aneuploid hybrids with specific chromosome substitutions was carried out at all stages of cytogenetic (metaphase I of meiosis) and molecular marker analysis (chromosome-specific SSR markers) at the plantlet stage. Cytogenetic screening is necessary to exclude the phenomenon of a “univalent shift,” where as chromosome-specific molecular markers confirm the presence of a specific chromosome substitution of *G. barbadense*. This analysis was noticeably easier, since all aneuploid lines in our collection were created in a common genotypic background of the highly inbred *G. hirsutum* L. L-458 line, which was created through multiple generations of self-pollination (F_20_) of cultivar 108-F.

### 2.2. Features of Crossing, Setting, and Germination of Hybrid Seeds

We crossed 30 monosomic and two monotelodisomic lines of *G. hirsutum* L. from the Cytogenetic Collection of Uzbekistan with the *G. barbadense* L. doubled haploid line Pima 3-79. The analysis of the crosses of the 30 aneuploid *G. hirsutum* L. lines with Pima 3-79 revealed significant differences between the lines. Thus, four monosomic lines with deficiencies in chromosome **2** (Mo11, Mo16, Mo19, and Mo93) were characterized by a decrease in the crossing rate with small ranges of variation between the lines (from 50 to 73.33%) (Figure 2). Four monosomic lines (Mo66, Mo69, Mo72, and Mo73) out of 13 lines with deficiencies in chromosome **4** were distinguished by a significant decrease in the crossing rate (from 16.66 to 25%); however, one line (Mo60) stood out as having the maximum crossability (100%) (Figure 2).Five monosomic lines (Mo13, Mo34, Mo67, Mo92, and Mo95) with deficiencies in chromosome **6** also showed a significant decrease in the crossing rate (from 26.38 to 66.67%). Of the remaining six monosomic lines, three lines (Mo48, Mo42, and Mo17) with deficiencies in chromosomes **18**, **21,** and **22** from the D_t_subgenome were characterized as having the largest decrease in the crossing rate (16.66, 16.67, and 14.29%, respectively) (Appendix A, Figure 2).

Two monotelodisomic lines (Telo12 and Mo21) with deficiencies in chromosome arms **6** and **11** differed significantly in their crossing rates (50 and 100%, respectively) with line Pima 3-79 (Appendix A, Figure 2).

There were large differences in the setting of F_0_ hybrid seeds obtained from crosses of aneuploid cotton lines with the line Pima 3-79, with the exception of one crossing combination (Mo48 × Pima 3-79).

All four monosomic lines with deficiencies in chromosome **2** were characterized by a decrease in the setting of F_0_ hybrid seeds (from 31.33 ± 5.09 to 60.71 ± 10.48%) compared with the control (73.83 ± 4.25) (Appendix A, Figure 3).

The setting of F_0_ hybrid seeds obtained from crosses of 13 monosomic lines with deficiencies in chromosome **4** showed a wider range of decreases (from 26.03 ± 5.14 to 61.76 ± 8.33%). Five monosomic lines with deficiencies in chromosome **6** were also distinguished by a strong decrease in the setting of hybrid seeds, but with a narrower range (from 18.64 ± 5.07 to 35.48 ± 4.96).

Of the remaining six monosomic lines (Mo27, Mo94, Mo56, Mo48, Mo42, and Mo17), one line (Mo17) with a deficiency in chromosome **22** from the D_t_subgenome was characterized as having the strongest decrease in seed setting (up to 29.41%) compared to the control (73.83 ± 4.25%) (Appendix A, Figure 3) and to other monosomic lines with deficiencies in chromosomes **7**, **12**, **17**, **18**, and **21**.

There was a large difference in the setting of F_0_ hybrid seeds obtained from crosses of two monotelodisomic lines with deficiencies in chromosome arms **6** and **11** (69.05 ± 7.13 and 16.00 ± 4.23%, respectively), which was explained by the arms belonging to different nonhomologous chromosomes.

Only the F_0_ hybrid seeds from the crosses of two monosomic lines with deficienciesin chromosome **2** (Mo19 and Mo93) showed differences in their reduced germination (57.14 and 73.33%) (Appendix A, Figure 4).

The germination capacity of F_0_ hybrid seeds obtained from crosses of one monosomic line (Mo31) with a deficiency in chromosome **4** showed the strongest decrease (21.05%), while in the other four variants of crosses (Mo58, Mo66, Mo69 and Mo89), a smaller decrease was observed (from 36.36 to 53.85%) (Figure 4). The germination capacity of F_0_ hybrid seeds obtained from crosses involving five monosomic lines with deficiencies in chromosome **6** was found to be very high (from 66.67 to 100%). The germination capacity of F_0_ hybrid seeds obtained from crosses of monosomic lines with deficiencies in chromosomes **7**, **12**, **17**, and **18** showed a slight decrease (up to 84.21%), while the germination of hybrid seeds from two lines with deficiencies in chromosomes **21** and **22** decreased to a higher degree (to 61.90 and 70.00%, respectively). The germination of hybrid seeds from crosses involving two monotelodisome lines with deficiencies in one arm of chromosomes **6** and **11** was high and did not differ significantly (Appendix A, Figure 4).

In general, the germination capacity of F_0_ hybrid seeds obtained from crosses of aneuploid *G. hirsutum* L. lines with the *G. barbadense* L. line Pima 3-79 was high compared to the hybrid seeds of the control (66.67%).

### 2.3. Cytogenetic Characteristics of the Monosomic F_1_ Hybrids

Aneuploid F_1_ hybrids were isolated in 27 interspecies F_1_ hybrid families based on phenotypes and analysis of meiotic metaphase I, and in five variants, only several hybrid families were studied due to the low frequency of the target univalent transfer in monosomic offspring. In two of the monosomic F_1_ families (Mo75 × Pima 3-79 and Mo95 × Pima 3-79), four monosomic hybrids were detected; in five monosomic F_1_ hybrid families (Mo58, Mo59, Mo89, Mo34, and Mo94), three monosomic hybrids were isolated; in eight monosomic F_1_ hybrid families (Mo16, Mo19, Mo93, Mo7, Mo31, Mo70, Mo56, and Mo17), two monosomic hybrids were detected; and in the remaining ten hybrid families (Mo11, Mo38, Mo60, Mo69, Mo13, Mo67, Mo92, Mo27, Mo48, and Mo42), one aneuploid F_1_ hybrid was obtained. These results suggested differences in the ease of monosomic detection in various hybrid backgrounds and/or differences in the maternal transmission rates for various monosomes. Deficiencies in one chromosome arm occurred in the progenies of two monotelodisomic F_1_ hybrids. In one of the aneuploid F_1_ hybrid families, two hybrid monotelodisomic plants were detected (Figure 5), but in another aneuploid F_1_ hybrid family, one hybrid monotelodisomic plant was detected (Appendix A).

Metaphase I analysis in 49 monosomic F_1_ hybrids revealed that 47 monosomic plants exhibited modal chromosome pairing of 25 bivalents and one univalent. A monosomic F_1_ hybrid (2_8_) from the Mo34 × Pima 3-79 family was characterized with the presence of additional univalents (1.09 ± 0.16 per cell). Another monosomic F_1_ hybrid (106_5_) from the Mo95 x Pima 3-79 family formed one quadrivalent in one PMC (0.07 ± 0.06 per cell) (Appendix A).

Analysis of the univalent size in monosomic interspecific F_1_ hybrid revealed large-sized univalent in four families with monosomy for chromosome **2** (Figure 6a) as well as in five families with monosomy for chromosome **6** (Figure 6b,c) and in one family with monosomy for chromosome **12** (Mo94 × Pima 3-79) (Figure 6d). F_1_ hybrids from 10 families with monosomy for chromosome **4** (Mo7 × Pima 3-79, Mo31 × Pima 3-79, Mo38 × Pima 3-79, Mo58 × Pima 3-79, Mo59 × Pima 3-79, Mo60 × Pima 3-79, Mo69 × Pima 3-79, Mo70 × Pima 3-79, Mo75 × Pima 3-79, and Mo89 × Pima 3-79) had average-sized univalents (Figure 7a,b), which confirmed that the univalents belong to the A_t_subgenome, and the “univalent shift” was absent. This phenomenon could lead to the production of substitution lines for target chromosomes.

The study of the size of univalents in three crosses with monosomy for chromosomes **17**, **21,** and **22** revealed a medium to small size of univalents (Figure 7c); the Mo48 × Pima 3-79 family, which had a monosomy for chromosome **18**, showed small-sized univalents (Figure 7d), which confirmed that the monosomes belong to the D_t_subgenome.

At the telophase stage, the frequency of normal tetrad formation was analysed. The meiotic index, which was originally proposed by Love R.M. [35] for the evaluation of meiosis in wheat, reports the normal tetrad percentage and is an indicator of meiotic stability. All F_1_ hybrids with monosomy had a meiotic index (over 90.00%) higher than control plants, indicating that univalent chromosomes were included in micropores. However, six monosomic F_1_ hybrids (Mo7 × Pima 3-79, Mo31 × Pima 3-79, Mo38 × Pima 3-79, Mo60 × Pima 3-79, Mo92 × Pima 3-79, and Mo94 × Pima 3-79) demonstrated an increase in tetrads with micronuclei from 1.15 ± 0.15% (Mo60 × Pima 3-79) to 2.03 ± 0.58% (Mo31 × Pima 3-79) in comparison with control plants (L-458 × Pima 3-79—0.36 ± 0.18%), which demonstrated disturbances in univalent disjunction and the formation of imbalanced gametes in them (Appendix A, Figure 8 and Figure 9).

Pollen viability was high in some monosomic F_1_ hybrids. However, most monosomic F_1_ plants exhibited reduced pollen viability. Nine monosomic F_1_ plants had great reductions in pollen viability (from 71.34 ± 1.28 to 79.89 ± 1.32%), but 30 monosomic F_1_ plants had small reductions in pollen viability (from 80.79 ± 1.26 to 89.98 ± 1.21%)**.** Early haplodeficient microspore abortion occurred prior to the pollen stage (Appendix A, Figure 10).

### 2.4. Molecular Marker Analysis of Unknown Monosomics

To map SSR loci to chromosomes, we screened monosomic F_1_ hybrid plants for the L-458 allele using labelled and/or unlabelled primers. For SSR loci located at sites other than the chromatin-deficient segment, the L-458 marker was present, and F_1_ hybrids exhibited a heterozygous phenotype. In comparison, if an SSR locus was located on the segment missing from the F_1_ hybrid plant, the electropherogram would not show the L-458 allele and exhibit a hemizygous pattern for the donor allele from *G. barbadense* Pima 3-79.

Our results showed that one monosomic F_1_ hybrid plant (Mo94 × Pima 3-79) deficient for a copy of an unknown chromosome showed the presence of only six *G. barbadense*-specific SSR marker bands (BNL1227, BNL1707, BNL3261, BNL3594, BNL3835, and BNL3886) and the corresponding absence of the respective L-458 allele. The results helped determine the chromosomal identities of monosome Mo94 based on known chromosomal locations of the respective SSR markers. Because the abovementioned SSR markers were previously assigned to chromosome **12** of the A_t_subgenome [36], the SSR-based results for Mo94 indicate that it is monosomic for chromosome **12** (Table 1; Figure 11).

SSR-based deficiency analysis of monosomic F_1_ hybrid plants (Mo56 × Pima 3-79) deficient for an unknown chromosome showed the presence of only *G. barbadense*-specific SSR marker bands for BNL1606, BNL2471, BNL2496, BNL3371, BNL3955, TMB0874, and TMB2018, with deficiencies in the respective L-458 alleles. As all of these SSR markers were previously assigned to chromosome **17** of the D_t_subgenome [36,37], the results indicated that Mo56 is monosomic for chromosome **17** (Figure 12).

Our results also showed that a monosomic F_1_ hybrid (Mo42 × Pima 3-79) deficient for an unknown chromosome only had one *G. barbadense*-specific SSR marker BNL1705 and was deficient in the respective L-458 allele. Our results indicated that the univalent in Mo42 is chromosome **21** of the D_t_subgenome [36], as this SSR marker has been assigned to this chromosome (Table 1) (Figure 13).

Line Mo17 was found to be monosomic for chromosome **22** of the D_t_subgenome based on the polymorphic BNL673 marker used here. Another previously identified marker, JESPR235, was also located on chromosome **20** or **22** of the D_t_subgenome [34]. These results suggested that line Mo17 is monosomic for chromosome **22** of the D_t_subgenome [36] (Table 1).

Telo12 was found to be monotelodisomic based on the chromosome-specific SSR markers BNL2884 and TMB0154 that were polymorphic in the monosomic F_1_ hybrids (Telo12 × Pima 3-79), and these markers were located on chromosome **6** of the A_t_subgenome [36,37] (Table 1).

Thus, we identified chromosomes for four monosomic lines and one monotelodisomic line by means of F_1_ aneuploid hybrids and chromosome-specific SSR markers that were previously assigned to specific cotton chromosomes based on deletion molecular analysis [38].

### 2.5. Cytological Identification and Numeration of Unknown Monosomes

In our collection, monosomes were identified and numerated by analysing meiotic metaphase I configurations of monosomic translocation heterozygous F_1_ hybrids, which were obtained by numerous sexual crosses between monosomic lines and tester-translocation lines from a cytogenetic collection from the USA.

Cytological analysis of seven hybrid variants involving seven tester translocation set lines (TT1L-7L, TT2R-8Rb, TT3L-6L, TT3R-5R, TT4R-15L, TT9R-25, and TT10R-11R) suggested that Mo94 was not monosomic for chromosome 1, 2, 3, 4, 5, 6, 7, 8, 9, 10, 11, 15, or 25 because the modal MI pairing configuration in the respective monosomic F_1_ hybrids included 23^II^ + 1^IV^ + 1^I^. In contrast, translocation line—TT11R-12L showed that Mo94 could be monosomic for chromosome 11 or 12 because in the monosomic F_1_ hybrid of Mo94 × TT11R-12L, the modal MI pairing configuration was 24^II^ + 1^III^. Molecular-marker data indicated that Mo94 must be monosomic for chromosome **12** of the A_t_subgenome.

Tests involving two translocation lines (TT7R-21R and TT20R-21L) showed that Mo42 could be monosomic for chromosome 7, 20, or 21, because in the two monosomic hybrids of—Mo42 x TT7R-21R and Mo42 x TT20R-21L, the modal MI pairing configurations were 24^II^ + 1^III^ (Figure 14). Since the monosomic line Mo42 shared a common chromosome **21** with the two translocation lines, the marker BNL1705 that was located on chromosome **21** of theD_t_subgenome could distinguish Mo42 from chromosome **21** of the D_t_subgenome.

Cytological tests of the four hybrid combinations of the four tester translocation lines (TT1L-7L, TT9R-25, TT11R-12L, and TT15R) suggested that Mo17 is not monosomic for chromosome 1, 7, 9, 11, 12, 15, 20, or 25 because the modal MI pairing configuration in the respective monosomic F_1_ hybrids included 23^II^ +1^IV^ +1^I^. In contrast, the test with translocation line TT20L-22R showed that Mo17 could be monosomic for chromosome 20 or 22, because in the monosomic F_1_ hybrid of Mo17 × TT20L-22R, the modal MI pairing configuration was 24^II^ + 1^III^. Molecular-marker (BNL673) data indicated that Mo17 must be monosomic for chromosome **22** of the D_t_subgenome.

Unfortunately, it is not currently possible to obtain cytogenetic confirmation of the homology of the univalent chromosome in the monosomic line Mo56 due to the small number of buds and the difficulty of detecting meiosis in the hybrid monosomic plant.

### 2.6. Some Features of the Newly Identified Monosomic Lines of *G. hirsutum* L.

Different cotton chromosomal deficiencies had a specific influence on plant morphology and other characteristics, such as bushes, flowers, and bolls. The initial primary monosomic plant of Mo94 with a deficiency in chromosome 12 was obtained in the third generation by pollination with irradiated pollen at a dose of 20 Gy. The monosomic line Mo94 is characterized by a large univalent size, a high meiotic index (95.73 ± 0.42), a small number of tetrads with micronuclei (0.52 ± 0.05), and reduced pollen fertility (76.82 ± 1.72%), as well as a low frequency of transmission to the progeny (18.18%) and a reduced frequency of transmission of n-1 gametes. This Mo94 line was determined to have typical phenotypic characteristics (Table 2; Figure 15) and a low seed setting (26.67 ± 3.44) compared to the original L-458 inbred line (89.81 ± 1.55). This decrease in seed setting occurred due to the presence of a large number of unfertilized ovules in the form motes in the monosomic bolls, which, when combined with a decrease in the number of seeds per boll (8.80 ± 4.27), led to a decrease in the size of the bolls.

The initial plant of the Mo56 monosomic cotton line with a deficiency in chromosome **17** was obtained by irradiation of the seeds of the L-458 line with thermal neutrons at a dose of 35 Gy. This line was characterized by a medium-small-sized univalent, a high meiotic index (97.48 ± 0.43), a small number of tetrads with micronuclei (1.22 ± 0.30%), and reduced pollen fertility (87.71 ± 0.99%), as well as a very low frequency of transmission to the progeny (7.69%), which significantly reduced the frequency of transmission of haplo-deficient gametes. Monosomic line Mo56 was distinguished by a reduced seed setting (56.55 ± 3.36) compared to the original inbred line L-458 (89.81 ± 1.55%) (Table 2, Figure 16). This decrease in seed setting occurred due to the presence of a large number of unfertilized ovules in the form of motes in the monosomic bolls, which, when combined with a decrease in the number of seeds per boll (12.00 ± 0.52), led to a decrease in the size of the bolls. Monosomic lines with a deficiency in chromosome **17** are characterized by complex morphobiological characteristics.

The initial plant of the Mo42 monosomic cotton line with a deficiency in chromosome **21** was obtained in the first generation by pollination with irradiated pollen from the L-458 line at a dose of 20 Gy. This line was characterized by a medium-small-sized univalent, a high meiotic index (95.87 ± 0.47), a small number of tetrads with micronuclei (0.45 ± 0.16%), and reduced pollen fertility (87.71 ± 1.53%), as well as a low frequency of transmission to the progeny (16.00%), which significantly reduced the frequency of transmission of haplo-deficient gametes. Monosomic line Mo42 was distinguished by a reduced seed setting (24.62 ± 3.72) compared to the original inbred line L-458 (89.81 ± 1.55%). This decrease in seed setting occurred due to the presence of a large number of unfertilized ovules in the form of motes in the monosomic bolls, which, combined with a decrease in the number of seeds per boll (6.60 ± 4.32), led to a decrease in the size of the bolls (Table 2; Figure 17).

The initial plant of the Mo17 monosomic cotton line with a deficiency in chromosome **22** was induced in the first generation by pollination with irradiated pollen from the L-458 line at a dose of 25 Gy. This line was characterized by a medium-small-sized univalent, a high meiotic index (96.68 ± 0.43), a small number of tetrads with micronuclei (1.90 ± 0.22%), and high pollen fertility (95.16 ± 0.49%), as well as a low frequency of transmission to the progeny (19.35%), which significantly reduced the frequency of transmission of haplo-deficient gametes. Monosomic line Mo17 was distinguished by a reduced seed setting (58.64 ± 3.32) compared to the original inbred line L-458 (89.81 ± 1.55%). This decrease in seed setting occurred due to the presence of a large number of unfertilized ovules in the form of motes in the monosomic bolls, which, combinedwith the decreased number of seeds per boll (12.90 ± 0.77), led to a decrease in the size of the bolls (Table 2; Figure 18).

The initial plant of the monotelodisome line, Telo12, which is deficiency an arm of chromosome **6**, was obtained in the second generation by pollination with irradiated pollen from the L-458 line at a dose of 20 Gy. This line was characterized by a high meiotic index (91.08 ± 1.38) and an increase in the percentage of tetrads with micronuclei (3.08 ± 0.89%), which disturbed monotelosome disjunction and resulted in imbalanced gametes. Telo12 had high pollen fertility (91.12 ± 1.80%) and a low frequency of transmission to the progeny (19.35%), which significantly reduced the frequency of transmission of haplo-deficient gametes. The monotelodisome line Telo12 was distinguished by a high seed setting rate (87.41 ± 2.69) compared to the original inbred line L-458 (89.81 ± 1.55%) and a decrease in the number of seeds per boll (26.00 ± 3.55) (Table 2; Figure 19).

### 2.7. Peculiarities of the Crossing, Setting, and Germination of BC_1_F_1_ Hybrid Seeds

The aneuploid lines of our collection were crossed with interspecific aneuploid F_1_ hybrids (Mo × Pima 3-79 or Telo× Pima 3-79). Most of the backcross hybrids were obtained from crosses of the original aneuploid lines with deficiencies in chromosomes **2**, **4**, **6**, **7**, **12**, **17**, **18**, **21**, **22**, telo **6**, and telo **11** with F_1_ hybrids (Mo × Pima 3-79) characterized as having a strong decrease in the crossing rate (up to 12.50%). Only three monosomic lines (Mo56, Mo42, and Mo17) with deficiencies in chromosomes **17**, **21**, and **22**, respectively, showed a strong increase in the crossing rate (up to 75, 75, and 50%, respectively) compared to the original F_1_ hybrids (Mo × Pima 3-79) (Figure 20). Five monosomic lines (Mo31, Mo60, Mo75, Mo27, and Mo21) showed a small decrease in the crossing rate (up to 50, 64.29, 60, 76.92, and 80%) compared to the F_1_ hybrids (Appendix A). Unfortunately, in the monosomic F_1_ hybrid (Mo13 × Pima 3-79) with a deficiency in chromosome **6**, for many crosses (25 crosses), not a single boll was set with the monosomic line; therefore, it is not possible to study these BC_1_F_1_ hybrids at present.

The setting of BC_1_F_1_ hybrid seeds obtained from crosses of aneuploid lines with the F_1_ hybrids (Mo × Pima 3-79 or Telo × Pima 3-79) was decreased in most lines. The exceptions included four monosomic lines (Mo16, Mo31, Mo67, and Mo27) with deficiencies in chromosomes **2**, **4**, **6,** and **7**, which, when backcrossed to the F_1_ hybrids, showed a small increase in the setting of hybrid seeds (Appendix A, Figure 21).

The germination capacity of the backcrossed hybrid seeds obtained from crosses of the original aneuploid lines with F_1_ hybrids (Mo × Pima 3-79 or Telo × Pima 3-79) showed a strong decrease (up to 37.50%) in only one variant F_1_BC_1_ (Mo92 × F_1_688_9_) with the substitution of chromosome **6**. In the remaining 13 variants, there was a slight decrease in the germination capacity of backcrossed seeds, and in seven variants, there was an increase in the germination capacity compared to the F_1_ hybrids. In general, the backcrossed F_1_BC_1_ seeds were characterized as having a high germination capacity (Appendix A, Figure 22).

### 2.8. Cytogenetic Characteristics of Aneuploid BC_1_F_1_ Hybrid Plants with Identified Monosomes

Among the resulting progeny, aneuploid BC_1_F_1_ hybrid plants were determined for 16 BC_1_F_1_ hybrid families based on hybrid phenotypes and meiotic metaphase I configuration analyses. Three monosomic plants were isolated from one aneuploid backcross BC_1_F_1_ family (Mo48); two hybrid monosomic plants were detected in each of six aneuploid backcross families (Mo60, Mo75, Mo34, Mo27, Mo94, and Mo17); and one hybrid monosomic plant was isolated from each of the remaining seven backcross families (Mo16, Mo38, Mo58, Mo59, Mo92, Mo56, and Mo42). Unfortunately, no monosomic plants were isolated from five aneuploid backcross families (Mo93, Mo7, Mo31, Mo67, and Mo95). These results suggested that there were differences in the transmission rate of monosomes in the various hybrid backgrounds and/or differences in the maternal transmission rates in different hybrid families. Deficiencies in one chromosome arm occurred in the progenies of two monotelodisomic BC_1_F_1_ hybrid plants. In one of the aneuploid BC_1_F_1_ families, three hybrid monotelodisomic plants were detected, but in another aneuploid backcross family, one hybrid monotelodisomic plant was isolated (Appendix A).

Meiotic metaphase I analysis of 14 monosomic BC_1_F_1_ plants revealed that 22 monosomes exhibited modal chromosome pairing with 25 bivalents and one univalent. One monosomic plant (110_3_) from BC_1_F_1_ (Mo17 × F_1_685_7_) formed a quadrivalent (0.38 ± 0.17 per cell) (Appendix A).

Analysis of the size of the monosomes in monosomic backcross BC_1_F_1_ plants revealed large-sized univalents in one family with the substitution of chromosome **2** (BC_1_F_1_ (Mo16 × F_1_98_6_)) and in two families with the substitution of chromosome **6** (BC_1_F_1_ (Mo34 × F_1_688_9_) and BC_1_F_1_ (Mo92 × F_1_539_5_)) (Figure 23a), as well as in one family with the substitution of chromosome **12** (BC_1_F_1_ (Mo94 × F_1_8_1_)) (Figure 23b). Monosomic BC_1_F_1_ hybrids of five families with the substitution of chromosome **4** (BC_1_F_1_ (Mo38 × F_1_690_11_), BC_1_F_1_ (Mo58 × F_1_530_3_), BC_1_F_1_ (Mo59 × F_1_531_8_), BC_1_F_1_ (Mo60 × F_1_694_5_), and BC_1_F_1_ (Mo75 × F_1_104_2_)) (Figure 23c–e), as well as with the substitution of chromosome **7** (BC_1_F_1_ (Mo27 × F_1_687_4_)) (Figure 24a), had medium-sized univalent that confirmed that the monosomes were from the A_t_subgenome and the absence of “univalents shift.”

The study of the size of univalents in backcross plants of three variants (BC_1_F_1_(Mo56 × F_1_4_17_), BC_1_F_1_(Mo42 × F_1_528_1_), and BC_1_F_1_(Mo17 × F_1_685_7_)) with the substitution of chromosomes **17**, **21**, and **22**, respectively, revealed medium-small-sized univalents (Figure 24c), and in backcross plants of one variant (BC_1_F_1_ (Mo48 × F_1_529_16_)) with substitution of chromosome **18**, the univalent had a small size (Figure 24b), which confirmed that the monosomes were from the D_t_subgenome.

All monosomic BC_1_F_1_ plants showed a higher meiotic index (more than 90%) than the control plants, which indicated that their univalent chromosomes underwent regular disjunction. However, five monosomic BC_1_F_1_ plants demonstrated an increase in the percentage of tetrads with micronuclei from 1.17 ± 0.26% (F_1_BC_1_ (Mo60 × F_1_694_5_) to 2.60 ± 0.81% (F_1_BC_1_ (Mo38 × F_1_690_11_)) in comparison with the control plants (L-458 × Pima 3-79—0.04 ± 0.04%), which demonstrated disturbances in monosome disjunction and the formation of imbalanced gametes in these BC_1_F_1_ hybrids. Moreover, two monotelodisomic BC_1_F_1_ plants from one family (F_1_BC_1_ (Mo21 × F_1_100_1_)) showed an increase in the percentage of tetrads with micronuclei from 2.46 ± 0.33% (292_1_) to 2.55 ± 0.67% (291_1_) in comparison with control plants (L-458 × Pima 3-79—0.04 ± 0.04%), which suggested disturbances in monotelodisome disjunction and the formation of imbalanced gametes in these BC_1_F_1_ hybrids (Appendix A, Figure 25).

Pollen viability after 2% acetocarmine staining was detected in aneuploid BC_1_F_1_ families. Most monosomic hybrid BC_1_F_1_ plants exhibited reduced pollen viability. Specifically, three monosomic hybrid BC_1_F_1_ plants had greater reductions in pollen viability (from 67.27 ± 1.45 to 78.60 ± 1.53%), but nine monosomic hybrid F_1_BC_1_ plants showed small reductions in pollen viability (from 80.30 ± 1.71 to 89.79 ± 0.94%) (Appendix A, Figure 26).

### 2.9. Identification of Univalent Chromosomes in Aneuploid F_1_BC_1_ Hybrids Obtained from Crosses between Aneuploid *G. hirsutum* L. Lines and Interspecific Aneuploid F_1_ Hybrids (Mo × Pima 3-79) Using Molecular Genetic Markers

Univalent chromosomes in aneuploid BC_1_F_1_ hybrids obtained from crosses of monosomic and monotelodisomic lines of *G. hirsutum* L. with interspecific aneuploid F_1_ hybrids (Mo × Pima 3-79) were identified using molecular genetic markers that were previously assigned to chromosomes.

To facilitate the detection of monosomic cytotypes and accelerate the production of backcross bolls in four BC_1_F_1_ hybrid families, molecular genetic analysis of hybrid plants at the plantlet stage was carried out before transplanting seedlings into greenhouse soil and carrying out cytogenetic analysis to identify monosomic forms (Table 3).

The results of this study showed that three backcross plantlets (922_2_, 922_8_, and 923_7_) were found in two hybrid backcross families (922 and 923) with the substitution of chromosome **2** (BC_1_F_1_(Mo16 × F_1_98_6_)); these plantlets were characterized by the presence of polymorphic alleles only from *G. barbadense*, while alleles of the *G. hirsutum* line L-458 were absent based on the localization of three chromosome-specific SSR markers: BNL834, TMB0471, and JESPR179 (Table 3). Since the previously identified markers were located on chromosome **2** of the A_t_subgenome [36,39], it can be assumed that the above hybrid plants have substitutions of this chromosome (Table 3; Figure 27). Additional cytogenetic analysis revealed a monosomic state in another backcross hybrid (923_8_), which has not been studied using molecular genetics due to poor development and few leaves present at the plantlet stage.

Four backcross plantlets were found (924_1_, 925_4_, 925_7_, and 925_11_) in two hybrid backcross families (924 and 925) with the substitution of chromosome **4** (BC_1_F_1_ (Mo31 × F_1_770_1_) and BC_1_F_1_ (Mo38 × F_1_690_11_)); these lines were characterized by the presence of polymorphic alleles only from *G. barbadense*, while alleles of the *G. hirsutum* line L-458 were absent based on the location of four chromosome-specific SSR markers: BNL2572, TMB0809, Gh117, and Gh107. Since the previously identified markers were located on chromosome **4** of the A_t_subgenome, we can assume that the above hybrid plantlets have substitutions of this chromosome (Table 3). Further cytogenetic analysis confirmed the monosomic state of only two backcross hybrids (925_4_ and 925_7_). In this regard, it is of great interest to use previously identified chromosome-specific SSR markers from cotton to test backcrossed plantlets for the early detection of monosomic cytotypes and their early involvement in subsequent backcrossing.

Confirmation of the substitution of other chromosomes was carried out in eight cytogenetically studied monosomic BC_1_F_1_ hybrids by common methods. The analysis of one monosomic hybrid (117_5_) from the variant BC_1_F_1_ (Mo60 × F_1_694_5_) with the substitution of chromosome **4** using molecular genetic markers showed the presence of polymorphic alleles only from *G. barbadense*, while the alleles of the *G. hirsutum* line L-458 were absent based on the chromosome-specific SSR markers Gh107, TMB0809, Gh117, CIR249, and JESPR234. Since the previously identified markers were located on chromosome **4** of the A_t_subgenome, it can be assumed that the substitution of this chromosome was confirmed in the studied monosomic hybrid (Table 4 and Table 5).

Additional molecular genetic analysis of the other two monosomic hybrids (298_2_ and 298_3_) in the BC_1_F_1_ family (Mo75 × F_1_104_2_) with the substitution of chromosome **4** revealed the presence of polymorphic alleles only from *G. barbadense*, while the alleles from the *G. hirsutum* line L-458 were absent based on the chromosome-specific SSR markers BNL2572, Gh107, TMB0809, Gh117, CIR249, and JESPR234. Since the previously listed markers were located on chromosome **4** of the A_t_subgenome [36,38], it can be assumed that the substitution of this chromosome 4 is confirmed in the above hybrid plants (Table 4 and Table 5).

A monosomic hybrid (293_3_) from the BC_1_F_1_ family (Mo34 × F_1_688_9_) was studied using molecular genetic markers; the results showed only polymorphic alleles from *G. barbadense*, while alleles from the *G. hirsutum* line L-458 were not found based on the chromosome-specific SSR markers BNL1440, BNL3650, BNL2884, BNL1064, BNL3359, TMB1277, TMB0154, TMB0853, TMB1538, Gh039, and Gh082. Since the previously identified markers were located on chromosome **6** of the A_t_subgenome, it can be assumed that the substitution of this chromosome is confirmed in the above hybrid plants (Table 4 and Table 5, Figure 28).

The molecular genetic analyses of another monosomic hybrid (1040_2_) in the BC_1_F_1_ family (Mo92 × F_1_539_5_) with the substitution of chromosome **6** showed the presence of polymorphic alleles only from *G. barbadense*, while the alleles from the *G. hirsutum* line L-458 were absent based on the chromosome-specific SSR markers BNL1440, BNL3650, BNL2884, BNL3359, TMB1277, TMB0154, TMB0853, TMB1538, Gh039, and Gh082. Since the previously identified markers were located on chromosome **6** of the A_t_subgenome, it can be assumed that the substitution of this chromosome wasconfirmed in the studied monosomic hybrid (Table 4 and Table 5; Figure 28).

The molecular genetic analysis of two monosomic hybrids (299_1_ and 299_2_) from the BC_1_F_1_ family (Mo94 × F_1_8_1_) revealed only polymorphic alleles from *G. barbadense*, while alleles from the *G. hirsutum* line L-458 were not detected based on the chromosome-specific SSR markers BNL1227 and BNL3835 in hybrid 299_1_ and BNL3594 in hybrid 299_2_. Since the previously identified markers were located on chromosome **12** of the A_t_subgenome [36], it can be assumed that the substitution of this chromosome was confirmed in the studied monosomic hybrids (Table 4 and Table 5).

The molecular genetic analysis of the monosomic hybrid (110_3_) from the BC_1_F_1_ family (Mo17 × F_1_685_7_) also showed the presence of only polymorphic alleles from *G. barbadense*, while the alleles from line L-458 were not detected based on the chromosome-specific SSR marker BNL673. Since this marker was previously localized to chromosome **22** of the D_t_subgenome [36], it can be assumed that the substitution of this chromosome was confirmed in the studied monosomic hybrid (Table 4 and Table 5).

### 2.10. Analysis of the Unique Morphological Characteristics of F_1_ and BC_1_F_1_ Aneuploid Hybrids in Comparison with the Original Aneuploid Lines

Analysis of the unique morphological characteristics of aneuploid lines and F_1_ and BC_1_F_1_ hybrids obtained from crosses of aneuploid lines acting as recurrent parents with Pima 3-79 and with interspecific aneuploid F_1_(Mo × Pima 3-79) hybrids was carried out, and the results were compared with those for the original monosomic lines.

Monosomic lines with a deficiency in chromosome **2** were distinguished by small narrow leaves, shortened sympodial branches, and small round bolls, while F_1_ hybrids showed an increase in leaves, flowers, and bolls. However, the monosomic BC_1_F_1_ hybrid showed a decrease in the plant growth and development rates, as well as a decrease in the size of the leaves, flowers, and bolls and the number of the bolls (Figure 29).

Monosomic lines with a deficiency in chromosome **4** were characterized by dense and vibrant plants, long leaf lobes, elongated bracts and pedicels, and ribbed bolls; all these features were preserved in monosomic F_1_ hybrids but with larger-sized peduncles and bolls, as well as an increase in the pubescence of the main stem. In monosomic BC_1_F_1_ hybrids with the substitution of chromosome **4**, dense foliage, densely pubescent stems with short hairs and medium-pubescent leaves, as well as medium-sized leaves and long pedicels (Figure 30), were found with the exception of the monosomic BC_1_F_1_ hybrid (Mo60 × F_1_694_5_), in which the peduncle length was short.

Monosomic lines with a deficiency in chromosome **6** were characterized by shortened sympodia, stiff stems, small leaves, small round bolls, and late flowering, while monosomic F_1_ hybrids had larger dark green leaves, longer sympodia, and larger bolls. In monosomic BC_1_F_1_ hybrid plants, a decrease in the size of leaves, flowers, and bolls, as well as shortening of the internodes, was observed (Figure 31).

Another monosomic line, Mo27, with a deficiency in chromosome **7** had characteristic features such as short sympodia, thick bracts and leaves, and small bolls. Monosomic F_1_ hybrids were characterized as having large leaves, flowers, and bolls, while monosomic BC_1_F_1_ hybrids were distinguished by their pale green leaves and short sympodia (Figure 32).

Monosomic line Mo94 with a deficiency in chromosome **12** showed a spreading form and a slightly leafy bush, shortened internodes, large leaves with slightly curved edges, and small bolls. Monosomic F_1_ hybrids were characterized as having the same traits, along with weak foliage, large bolls, and a decrease in the number of bolls, while monosomic BC_1_F_1_ hybrid plants were more densely leafed, had elongated sympodial branches, and had large flowers and bolls (Figure 33).

The monosomic line Mo56 with a deficiency in chromosome **17** had characteristic features such as a compact bush, dense foliage, small leaves, short stalks, small bolls, abundant budding, and a weak boll set. These traits were retained in monosomic F_1_ hybrids, along with larger bolls and a decrease in the number of bolls. In addition, the monosomic BC_1_F_1_ hybrid had dense foliage and an increase in the number of bolls (Figure 34).

The monosomic line Mo48 with a deficiency in chromosome **18** had small leaves, a long column and stigma, and short sympodia. These traits were retained in monosomic F_1_ hybrids; in addition, medium-sized leaves and bolls were formed. Monosomic BC_1_F_1_ hybrids showed weak foliage, three-lobed leaves, and pubescent petioles and bracts (Figure 35).

Monosomic line Mo42 with a deficiency in chromosome **21** had characteristic features such as a compact bush, weak foliage, small three-lobed leaves, shortened sympodial branches, and small ovoid bolls. These traits were preserved in the F_1_ monosomic hybrid; in addition, large leaves and bolls were observed. In monosomic BC_1_F_1_ hybrids, the leaf shape was three-lobed, and the leaves and bolls were smaller than those of the monosomic F_1_ hybrids (Figure 36).

Another monosomic line, Mo17, with a deficiency in chromosome **22** was characterized as having vibrant plants, elongated leaf lobes with a slight curving of the edges upward, long bracts and pedicels, elongated ribbed bolls, and weak budding. These traits were retained in the F_1_ monosomic hybrids, along with weak foliage. Monosomic BC_1_F_1_ hybrids differed in their dark green leaves and ovoid bolls of a medium size (Figure 37).

The monotelodisome line Telo12 with a deficiency in an arm of chromosome **6** had characteristic features such as a compact bush, dense foliage, small leaves, and small, spherical bolls with elongated noses. These traits were preserved in the F_1_ monotelodisome hybrids along with large leaves and long cuttings in leaves, while in BC_1_F_1_ hybrids, the bolls were larger and more ovoid (Figure 38).

Another monotelodisome line, Mo21, with a deficiency in an arm of chromosome **11** was characterized by a spreading bush, dense foliage, dark green leaves of a medium size that were whole-edged and three-lobed, small bolls, and a very low seed setting. The monotelodisome F_1_ hybrids had dark green, two- to three-lobed leaves, and spherical leaves with shallow-nosed bolls, while the BC_1_F_1_ monotelodisome hybrids presented leaves of a large size with long petioles and egg-shaped bolls (Figure 39).

Since there are no detailed morphobiological descriptions of F_1_ and BC_1_F_1_ hybrids with substitutions on chromosomes **2**, **4**, **6**, **7**, and **12** and the arms of chromosomes **6** and **11** of the A_t_subgenome and chromosomes **17**, **18**, **21**, and **22** of the D_t_subgenome in *G. barbadense*, direct comparison of our data with that from the literature is not possible.

Thus, some aneuploid BC_1_F_1_ hybrids obtained from crosses of monosomic and monotelodisome lines with interspecific aneuploid F_1_ hybrids with substitutions of chromosomes **2**, **4**, **6**, **7**, and **12** and the arms of chromosomes **6** and **11** of the A_t_subgenome and chromosomes **17**, **18**, **21**, and **22** of the D_t_subgenome in *G. barbadense* showed some distinctive features. Such features included dense foliage, densely pubescent stems with short hairs, and medium-pubescent leaves in the monosomic BC_1_F_1_ hybrids (Mo58 × F_1_530_3_ and Mo75 × F_1_104_2_) (115_1_ and 298_2_, respectively), as well as medium-sized leaves and long pedicels (from 3 to 5 cm), with the exception of the monosomic BC_1_F_1_ hybrid (Mo60 × F_1_694_5_), in which the peduncle length was short due the substitution of chromosome **4**. Monosomic BC_1_F_1_ hybrids with the substitution of chromosome **6** were distinguished by a compact bush, very dense foliage, shortened sympodial branches, thicker and darker green leaves, short peduncles, and spherical bolls, while monotelodisome hybrids with the substitution of an arm of chromosome **6** showed larger leaves with an increase in the number of leaves with long petioles (up to 15 cm), as well as egg-shaped bolls. Monosomic BC_1_F_1_ hybrids with the substitution of chromosome **7** were distinguished by pale green-coloured leaves, a three-lobed and finger-lobed leaf shape, and a smaller number of teeth on the bracts (up to eight), while monosomic BC_1_F_1_ hybrids with the substitution of chromosome **18** were distinguished by a three-lobed leaf shape, shortening of the length of the petiole and dense pubescence, shortened stigma protrusion, and spherical bolls. Monotelodisome BC_1_F_1_ hybrids with the substitution of an arm of chromosome arm **11** were distinguished by thick, two- to three-lobed medium-sized leaves, a decrease in the number of teeth on the bracts, and spherical-shaped bolls. All aneuploid BC_1_F_1_ hybrids with substitutions of specific chromosomes or their arms were found to lack an anthocyanin spot on the petals. Therefore, it can be assumed that the persistence of changes in some traits in aneuploid BC_1_F_1_ hybrids in future backcross generations will indicate the connection of these changes with substituted chromosomes or specific chromosome arms.

## 3. Discussion

The creation of introgression lines can contribute to the expansion of genetic diversity, the decrease in genetic vulnerability, the increase in resistance to various pathogens, and the improvement in breeding indicators. In addition, obtaining stable lines containing the substitution of specific chromosomes allows the introgression of genetic material associated with valuable traits [41]. The method used to develop chromosome-substitution lines in cotton has been discussed in numerous studies [22]. The strategy is based on differences in the transmission rate between mega- and microgametophytes, that is, between the egg (seed or female parent) and the pollen (male parent) in cotton [42]. The transmission of hypoaneuploidy in cotton through eggs is successful (up to 50%) for most chromosomes or chromosome arms, while transmission through pollen is rare or non-existent across all chromosomes and most large segmental deletions (for example, telosomes) [43].

Later, S. Saha et al. [23] described a method for obtaining chromosome-substitution lines in cotton that involved crossing the *G. barbadense* line Pima 3-79, as the male parent, and a monosomic or monotelodisome line of *G. hirsutum* as the female parent. Furthermore, interspecific F_1_ progeny were screened phenotypically, cytogenetically, and in some cases using molecular marker analysis to identify plants deficient in a chromosome or chromosome arm. The identified interspecific aneuploid (BC_0_F_1_) plant was subsequently used as the male parent when backcrossed with a recurrent aneuploid plant. This procedure was repeated until the fifth backcross. Then, one euploid BC_5_F_1_ plant was self-pollinated to identify each substituted BC_5_F_1_S_1_ line in which the (CS-B) chromosomes (or segments of the chromosome arms) of *G. hirsutum* were substituted with the corresponding pair from *G. barbadense*.

In this study, a scheme for the creation of chromosome-substituted cotton lines was developed, according to which all stages of hybridization and backcrossing were accompanied by double screening of specific chromosome substitutions using cytogenetic and molecular markers (SSRs) at early stages of development (plantlets). Here, the absence of amplification of an SSR locus that is associated with a specific chromosome in *G. hirsutum*, but the presence of markers from *G. barbadense,* was considered as a substitution of the homeologous chromosome of the first species with that of the second species. This scheme makes it possible to significantly accelerate the process of creating lines with substitutions of specific chromosomes and to control a possible “univalent shift.” The retarded development of monosomic plants and late budding and flowering in comparison with disomic siblings can delay backcrossing. After the early detection of monosomic cytotypes using the above markers at the plantlet stage, it becomes possible to quickly backcross hypoaneuploid hybrids at an earlier stage of development, even before their monosomic status is confirmed by cytogenetic analysis.

It is known that some previously obtained chromosome substitution lines from the USA collection did not receive further molecular-genetic confirmation (CS-B05sh, CS-B06, CS-B07, and CS-B15sh), and one of the DNA reserves that was previously considered to have arisen from a *G. barbadense* plant hemizygous for Chr-17 was actually hemizygous for Chr-11 [36,40,44]. It can be assumed that a “univalent shift” occurred during backcrossing, which led to the production of such lines.

In our work, a strong decrease in cross-breeding and the setting of hybrid seeds was revealed during the hybridization of aneuploid lines of *G. hirsutum* L. with Pima 3-79 of *G. barbadense* L., since many monosomic lines in our collection were initially characterized by a reduced seed set, as well as a small number of seeds per boll. It is known that a reduction in the size, number, or curvature of the boll due to a limitation in seed productivity and the presence of a large number of motes served as a marker for cytological aberrations in cotton [45].

The study of the germination of hybrid seeds obtained from crosses between aneuploid lines of cotton and the donor *G. barbadense* L. line Pima 3-79 revealed a significant decrease in germination capacity in only one variant (Mo19) with a deficiency in chromosome **2** and four lines (Mo31, Mo66, Mo69, and Mo89) with a deficiency in chromosome **4**. In the remaining variants, there was a slight decrease compared with the control. The germination of interspecific hybrid seeds is one of the most important indicators when working with aneuploid plants, since such plants have reduced growth and development rates, and the seeds of hypoaneuploid plants germinate much later than those of disomic sibs. Thus, nonsimultaneous germination of seeds from disomic cytotypes and cytotypes with haplo-deficiency was observed.

A comparative analysis of chromosome pairing in monosomes of F_1_ hybrids obtained from crosses of 24 monosomic and two monotelodisome lines with the donor line Pima 3-79 revealed normal pairing of chromosomes with the formation of 25 bivalents and one univalent of different sizes in all studied PMCs. This result indicated the absence of the formation of additional univalents and served as a guarantee of the absence of a “univalent shift” in the tested monosomic plants resulting from a series of backcrosses. It is known that univalent chromosomes in three cotton monosomes (Mono12, Mono22, and Mono25) underwent misdivision much more often than other chromosomes [42,45,46], which indicated their instability.

Our detection of a high meiotic index and the presence of a small number of tetrads with micronuclei in the analysis of the microspore stage in monosomic F_1_ hybrids indirectly indicated the regularity of meiotic division and confirmed the normal pairing of chromosomes in monosomic plants. The detection of some decrease in pollen fertility in monosomic hybrid F_1_ plants confirmed the existence of specific differences between them. In general, the regular pairing of chromosomes in interspecific aneuploid F_1_ hybrids, a high meiotic index, and a partial decrease in pollen fertility indicate latent structural variability that is not recorded at the metaphase I stage of meiosis.

The creation of a database of molecular markers (SSRs) and the assignment of such markers to chromosomes [36,37,38,39,47,48,49] made it possible to successfully use these tools in experiments to identify specific chromosomes of cotton. The results reported here clearly demonstrated that the use of molecular deletion analysis [38] to localize previously assigned chromosome-specific SSR markers made it possible to identify unknown univalent chromosomes in four new monosomic lines in our collection (Mo94, Mo56, Mo42, and Mo17). These lines have a deficiency in chromosome **12** from the A_t_subgenome and chromosomes **17**, **21**, and **22** from the D_t_subgenome, respectively; the monotelodisome line Telo12 has a deficiency in an arm of chromosome **6** from the A_t_subgenome.

Due to the small size, large number, and lack of differential banding, cotton chromosomes, similar to the chromosomes of maize [50], barley [51], peas [52], rye [53], and tomato [54], are identified using translocation test lines created by Brown M.S. [55]. The identification of cotton monosomes by the size of the univalents allows only conditionally attributing them to subgenomes, since it is known that the chromosomes of the A_h_subgenome are almost twice as long as those of the D_h_subgenome, but some of the smallest chromosomes of the A_h_subgenome are not much larger than the largest chromosomes of the D_h_subgenome; however, medium-sized chromosomes generally belong to the A_h_subgenome [56].

The principle by which the identification of chromosomes using translocations was carried out has previously been described [57]. In cotton, the detection of quadrivalent and univalent meiosis in the PMC in metaphase I in hybrid translocation monosomic plants indicates the nonhomology of chromosomes involved in translocation and in univalent chromosomes [45]. When trivalents are found in PMCs of hybrid translocation monosomic plants, this result indicates the homology of the univalent and one of the chromosomes in the translocation. To determine which chromosome in the translocation is homologous to the univalent chromosome, crosses of this monosomic plant with other translocation lines were carried out, in which one of the chromosomes in translocations was the same as that of the original line. An analysis of chromosome associations in monosomic hybrids makes it possible to identify a univalent chromosome as a specific chromosome of a set [42].

The results of the cytological tests of hybrids obtained from crosses of three monosomic lines with translocation test lines (F_1_Mo94 × TT11R-12L, F_1_Mo42 × TT7R-21R and F_1_Mo42 × TT20R-21L, and F_1_Mo17 × TT20L-22R) helped identify the trivalent chromosome associations of previously unknown chromosomes, such as chromosome **12** of the A_t_subgenome and chromosomes **21** and **22** of the D_t_subgenome of cotton. In general, the use of chromosome-specific markers and translocation test lines greatly facilitated the identification of unknown monosomes in monosomic lines in our collection.

These monosomic cotton lines are characterized by complex phenotypic traits due to the deficiency in specific chromosomes. Since the monosomic lines in our collection were created in a single genotypic background, that of the L-458 line, obtained on the basis of the 108-F cultivar, and the monosomic lines of the collection from the USA have a different genetic background based on the Deltapine 14 variety, with subsequent backcrossing with the Texas Marker 1 line; the differences between the monosomic lines of different collections are acceptable. Backcrossing of monosomic cotton lines with the initial F_1_ hybrids (Mo × Pima 3-79) led to a strong decrease in the crossing rate in many variants, with the exception of three variants involving the monosomic lines Mo56, Mo42, and Mo17 with deficiencies in chromosomes **17**, **21**, and **22**. Studying the setting of F_1_BC_1_ hybrid seeds also revealed a general decrease in the setting rate in 17 variants of crosses compared to the F_1_ hybrids (Mo × Pima 3-79), with the exception of variants involving four lines (Mo16, Mo31, Mo67, and Mo27). In general, the seed setting in F_1_ crosses was significantly higher than that in backcrosses. It was previously noted that the difference in the number of motes in disomic and monosomic plants is sufficient to reveal the difference between monosomic individuals and normal sibs [45].

The obtained results showed that it is important to consider the crossing and setting of hybrid seeds in the process of creating substitution lines, since for unknown reasons, F_1_BC_1_ hybrids have a strong decrease in these indicators, despite the regular pairing of chromosomes and the relatively high pollen fertility of F_1_ hybrids. This phenomenon was clearly demonstrated by monosomic line Mo13 with a deficiency in chromosome **6**, which did not produce a single backcrossed boll when crossed with the monosomic interspecific F_1_ hybrid. Apparently, the decrease in the setting of hybrid BC_1_F_1_ backcross seeds was due to the specific features of those hybrids, along with the hybridity of the material and the low seed set in many of the original lines with haplo-deficiency.

Unfortunately, we were unable to compare our data with the results of American studies, since no data on the cytogenetic characteristics of hybrids were published when chromosome-substitution lines were created via backcrossing. Therefore, in future studies, how these indicators change with an increase in backcrossing generations will be examined.

A decrease in the germination of BC_1_F_1_ hybrid seeds compared to the initial F_1_ hybrids (Mo × Pima 3-79), with the exception of six BC_1_F_1_ families involving Mo7, Mo31, Mo58, Mo59, Mo34, and Mo95, could occur as a result of the effect of alien substitutions of various nonhomologous chromosomes and the reduced germination rates of backcross seeds with haplo-deficiencies.

It was found that the karyotypic transformations of plants during backcrossing of the original cotton lines with the pollen from F_1_ hybrids were not accompanied by any visible disturbances in meiotic divisions, since a comparative analysis of chromosome pairing in 21 hybrid monosomic BC_1_F_1_ plants with substitutions of chromosomes **2**, **4**, **6**, **7**, and **12** from the A_t_subgenome and chromosomes **17**, **18**, **21,** and **22** from the D_t_subgenome revealed normal chromosome pairing. In addition, the detection of a high meiotic index and a small number of tetrads with micronuclei served as an indirect confirmation of the regularity of meiosis and the absence of the appearance of additional univalents. However, the detection of a significant decrease in pollen fertility in monosomic hybrids with the substitution of specific chromosomes in different BC_1_F_1_ families indicated the existence of specific differences in pollen fertility in these plants. It is known that the microscopic assessment of pollen fertility after staining the progeny of cotton monosomes with acetocarmine is not entirely convincing as a method for separating monosomic and disomic plants due to the abortion of unbalanced microspores in early development [45]. A strong decrease in pollen fertility combined with a high meiotic index occurred due to the formation of gametes with haplo-deficiencies, which looked like normal tetrads at the microspore stage.

In our study, hybrid plants with haplo-deficiencies were not found in all BC_1_F_1_ hybrid families, since it was not possible to detect monosomic plants in five of them (Mo7, Mo31, Mo67, Mo93, and Mo95) for various reasons. Success in the transmission of the monosomes was highly dependent on the rate of reproduction in the hybrid offspring. It is known that there are differences between specific chromosomes of the cotton genome in the viability of the haplo-deficient gametes produced [45]. In addition, it was noted that the transfer of the monosomic state in cotton monosomes is limited in all by ovules, and the low frequency of transmission of the monosomes indicates a low viability of gametes with haplodeficiency [42]. The data presented in this article suggest that the rate of transmission of haplo-deficiencies in cotton hybrids depends on the specificity of the monosome and the number of hybrid plants.

For molecular genetic confirmation of chromosome substitutions, accelerated detection of monosomic cytotypes, and rapid backcrossing of plants with haplo-deficiencies, chromosomal assignment of molecular markers was carried out at the plantlet stage (4–5 true leaves) for the first time in cotton plants with alien chromosome substitutions. The identification of seven monosomic cytotypes at the plantlet stage in four families with substitutions of chromosomes **2** and **4** using markers previously assigned to chromosomes is of great importance for identifying hybrids with alien chromosome substitutions. This method significantly speeds up the analysis of hybrid progeny and contributes to the detection of plants with haplo-deficiencies, which leads to an acceleration of backcrossing and the production of hybrid bolls. Confirmation of substitutions of alien chromosomes in other BC_1_F_1_ families, carried out in our study by the common molecular genetic analysis, based on six previously identified monosomic hybrids (Mo60, Mo75, Mo34, Mo92, Mo94, and Mo17), serves as the basis for further research on obtaining chromosome-substitution cotton lines. However, such studies require much more time than those using BC_1_F_1_ monosomes that were identified at the plantlet stage.

Comparative analysis of the morphobiological characteristics of monosomic F_1_ and BC_1_F_1_ hybrids in comparison with the original monosomic lines showed that monosomic hybrids of the first generation were more similar to the donor line, while BC_1_F_1_ hybrids with haplodeficiency shared more characteristics with the recurrent parent and presented previously unidentified characteristics. These new markers included strong pubescence of the main stem in the BC_1_F_1_ hybrid—with substitution of chromosome **4**, pale green coloured leaves in the hybrid—with substitution of chromosome **7**, dense foliage and large bolls in the hybrid—with substitution of chromosome **12**, pubescence of the bracts and peduncles in the hybrid—with substitution of chromosome **18**, dark green leaf colour in the hybrid—with the substitution of chromosome **22**, lengthening of the cuttings and an increase in the size of the bolls in the hybrid—with the substitution of an arm of chromosome **6**, and dark green leaves and two-three-lobed leaves in the hybrid—with the substitution of an arm of chromosome **11**.

The results reported in recent years have clearly demonstrated the associations of substituted cotton chromosomes with various traits: CS-B04—higher fibre length; CS-B06—higher fibre yield; CS-B07—higher mass of one boll and micronaires; CS-B16—reduced seed yield, fibre yield, and fibre length but increased fibre yield and weight per boll; CS-B17—increased fibre yield, boll mass, and fibre elongation but reduced fibre yield, micronair, and fibre length; CS-B18—reduced seed and fibre yield and weight of one boll, but higher fibre and micronair yield; and CS-B25—significant association with all traits, except for seed yield [24].

However, 103 SSR primers were not associated with specific chromosomes due to the absence of hypoaneuploid lines for some chromosomes in the genome. To date, for five chromosomes in the genome (**5**, **8**, **14**, **15**, and **22**), no monosomic plants have been obtained, while for three chromosomes (**13**, **19**, and **24**), no deficiencies have been found in either specific chromosomes or specific chromosome arms [40].

An analysis of the studies including the association of markers assigned to chromosome **6** once again revealed inconsistencies between different studies, which raised the suspicion that the origin of some aneuploids may contribute most to the unexpected shortcomings of such studies [36]. Later, the chromosome-substitution line CS-B06 was found not have a substitution for all of chromosome **6** from the A_t_subgenome, since extensive molecular genetic work with five SSR loci specific for chromosome **6** revealed the amplification of the Upland-type alleles, not the Pima 3-79 alleles, which collectively suggests that this lineage was missynthesized and/or not inbred [44].

In the future, new chromosome substitution cotton lines will serve as the basis for molecular genetic studies and chromosome localization of new molecular markers, as well as improving the quality of cotton fibre by crossing these lines with cultivated varieties and enriching them with loci responsible for improving the quality of the fibre.

## 4. Materials and Methods

### 4.1. Plant Material

Monosomic lines from the Cytogenetic Collection of Uzbekistan were developed in the common genetic background of the highly inbred *G. hirsutum* line L-458, which was created through multiple generations of self-pollination (F_20_) of the cultivar 108-F. Because all 95 primary monosomic plants in the collection were isolated from a common genetic background, the differences observed among them can be attributed to differences in their monosomic states [28,30]. The irradiation of seeds by thermal neutrons or gamma-irradiation of pollen gave rise to most (76) of the monosomic plants. Other monosomic plants were detected among progenies of desynaptic plants (17) and translocation heterozygous plants (2). The primary monosomic plants were numbered according to their order of detection (Mo1-Mo95).

The *G. barbadense* line Pima 3-79 is not photoperiod sensitive and is highly homozygous, as it originates from a doubled haploid [42]. It was used extensively as a parent in genetic and genomic studies [58] and as a donor parent of substituted chromosome (CS) or chromosome segments from *G. barbadense* in our study.

A well-defined tester set of translocation lines from the Cytogenetic Collection of the USA was kindly provided by Prof. D.M. Stelly through the USDA–Uzbekistan cotton germplasm exchange program. All plant materials were vegetatively maintained in the greenhouse of the National University of Uzbekistan.

### 4.2. Analyses of the Crossing, Setting, and Germination of Cotton Hybrid Seed

Determination of the crossing rate of monosomic lines of *G. hirsutum* L. with the *G. barbadense* L. donor line Pima 3-79 was carried out by calculating the number of bolls set from the number of crosses performed in percent. Similarly, the determination of the setting rate of hybrid seeds obtained from crosses was carried out by determining the number of full-fledged seeds from the total number of matured, unripe seeds, and unfertilized eggs in the form of beetles in percent. The germination capacity of hybrid seeds was calculated by counting the number of germinated seeds out of the number of seeds sown in percent. Statistical processing of the data obtained was carried out by determining the error of the sample mean according to a previously described formula [59].

### 4.3. Cytological Analyses

For studies of meiotic chromosome pairing at metaphase I (MI) and sporad normality, floral buds were collected in the morning and fixed in a solution of 96% alcohol and acetic acid (7:3) after the removal of the calyx and corolla. Buds were kept at room temperature for 3 days, immersed in fresh fixative, and stored at 4 °C. For cytological preparations, buds were rinsed in tap water before being examined for meiotic MI chromosome configurations in microsporocytes, commonly known as “pollen mother cells” (PMCs), using the iron acetocarmine squash technique. Analyses of hybrid plant chromosomes were carried out on the basis of MI configurations. The development of F_1_ and BC_1_F_1_ hybrid plant PMCs was assessed based on the cytological features observed at the tetrad stage. To analyse the stage of meiotic sporads, several buds from each hybrid variant were analysed, and the meiotic index (Mi) [35] was calculated as the percentage of normal tetrads of the total number of sporads: $\mathrm{Mi}=\frac{X}{N}\;\times \;100\%$, where X is the number of normal tetrads, and *N* is the total number of sporades. Pollen viability was estimated as the percentage of mature pollen grains that were stained with acetocarmine. All cytological observations were made using Biomed (Leica, Heerbrugg, Switzerland) and Axioskop A1 (Carl Zeiss, Oberkochen, Germany) microscopes. Microphotography was performed using an AxioCamERc5s digital camera.

### 4.4. Identification of Monosomic F_1_ Hybrid Translocation Heterozygotes

Monosomes were identified using translocation tests. For this purpose, the monosomic (2n = 51) lines from the Cytogenetic Collection of Uzbekistan were crossed as seed parents with translocation lines (2n = 52) of the tester set from the Cotton Cytogenetic Collection of the USA provided by Texas A&M AgriLife Research. Floral buds of F_1_ hybrid progeny were analysed to identify monosomic plants (2n = 51) and those heterozygous for the respective translocation, that is, monosomic translocation heterozygotes. To reveal the “critical configurations” and detect common chromosomes among the chromosomes involved in interchanges with monosomes, the meiotic MI configurations were analysed in monosomic translocation heterozygous F_1_ hybrids. Progeny with a 23^II^ + 1^IV^ + 1^I^ modality were interpreted as having independence between chromosomes affected by monosomy and translocation, whereas progeny with a 24^II^ + 1^III^ modality were interpreted as evidence of association between the monosome and one of the two translocated chromosomes. It was generally presumed that the monosomic states of the parents and progeny were similar and that a “univalent shift” did not occur.

### 4.5. DNA Extraction and Genotyping

Genomic DNA was extracted from young leaf samples of cytogenetically identified monosomic F_1_ and BC_1_F_1_ hybrids and young hybrid plants (plantlets) using the CTAB method [25]. Extracted genomic DNA was checked by 0.9% agarose electrophoresis, and DNA concentrations were diluted to a working concentration in 15 μL of solution based on 25 ng/μL Hind III-digested λ-phage DNA. PCR amplification was carried out in a 10 μL reaction mix containing 1.0 μL 10× PCR buffer (with 25 mM MgCl_2_), 0.2 μL BSA, 0.08 μL dNTPs (25 mM), 0.2 μL primers, 0.1 μL Taq-polymerase, and 2 μL DNA template. PCR runs were conducted with an initial denaturation of DNA at 94 °C for 2 min, followed by 35 cycles of 94 °C (step 1) for 20 s, 55 °C (step 2) for 30 s, and 72 °C (step 3) for 50 s. After 35 cycles, the extension temperature of 72 °C was held for 7 min. The PCR products were visualized in a 3.5% high-resolution agarose gel, stained with ethidium bromide and photodocumented using an Alpha Imager (Innotech Inc., Airport Road Roanoke, Roanoke, VA, USA) gel documentation system.

Chromosome-specific SSR primer pairs were collected according to recently published genetic-mapping studies [36,60]. For each chromosome, we chose four loci that were polymorphic between L-458 (*G. hirsutum*) and Pima 3-79 (*G. barbadense*). Documented electropherogram results for SSRs were scored as “a/b/h,” where the “a” locus was similar to recipient L-458; the “b” locus was similar to donor line 3-79; and the “h” genotype was similar to the normal F_1_ hybrid. To identify the chromosome deficiencies in a given F_1_ monosomic cotton hybrid, the SSRs on that chromosome were expected to exhibit the “b” genotype, that is, a deficiency in the *G. hirsutum* (maternal) allele and the presence of only the *G. barbadense* (paternal) allele [61].

## 5. Conclusions

In this article, a scheme for creating chromosome-substitution lines in cotton was developed, all stages of which are accompanied by double screening of specific chromosome substitutions using cytogenetic and molecular genetic markers (SSRs) to avoid a “univalent shift” during the backcrossing process.

A decrease in the crossing and setting rates of interspecific F_1_ and BC_1_F_1_ hybrid seeds was established due to the interspecific features of the crossed species and the specificity of the deficiencies in specific chromosomes, while the decrease in the germination capacity of hybrid seeds in some crosses can be explained by the reduced germination rates and the development of haplo-deficient hybrid plants. In general, the comparison of these indicators suggested that these values were significantly higher in the primary crosses than those in the BC_1_F_1_ backcross hybrids.

Our observations of the behaviour of chromosomes during meiosis and the detection of chromosome pairing, normal for monosomic cotton plants, without the formation of additional univalents served as a guarantee of the absence of a “univalent shift.” A partial decrease in pollen fertility, which increased in backcross hybrids despite a high meiotic index and a small number of tetrads with micronuclei, indicated hidden structural variability in the *G. barbadense* and *G*. *hirsutum* genomes that was not recorded at the metaphase I stage of meiosis.

The identification of univalent chromosomes in four new monosomic lines (Mo94, Mo56, Mo42, and Mo17) with deficiencies in chromosome **12** and an arm of chromosome **6**, the A_t_subgenome, or chromosomes **17**, **21**, and **22** of the D_t_subgenome using translocation and previously assigned chromosome-specific SSR markers enriched the collection of hypoaneuploid cotton lines. For each such line, an analysis of the suite of phenotypic traits that are altered due to the deficiencies in the above chromosomes was carried out. The study of morphobiological features revealed the similarity between the aneuploid F_1_ hybrids and the donor line, while backcross aneuploid BC_1_F_1_ hybrids shared a more similar appearance with the recurrent parent, along with other previously undetected characteristics. Thus, the features of new chromosome-substitution lines were shown using various methods, and these lines may serve as sources of introgression of new traits and properties from *G. barbadense/G. hirsutum*.

## Figures and Tables

**Figure 1 plants-11-00542-f001:**
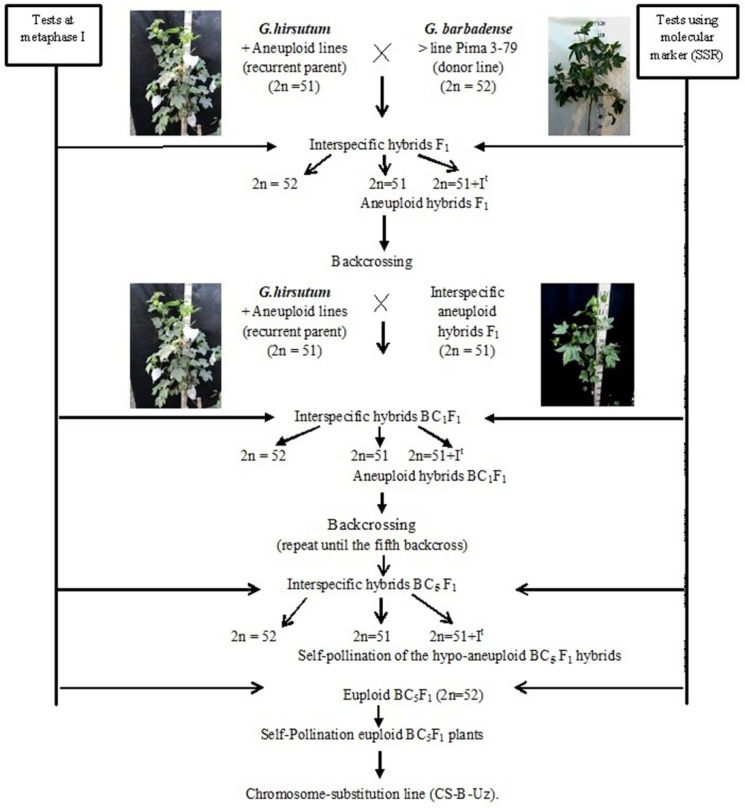
Development of cotton chromosome-substitution lines. The scheme shows the different steps for obtaining cotton chromosome-substitution lines, including cytogenetic analysis and the use of molecular markers (SSRs).

**Figure 2 plants-11-00542-f002:**
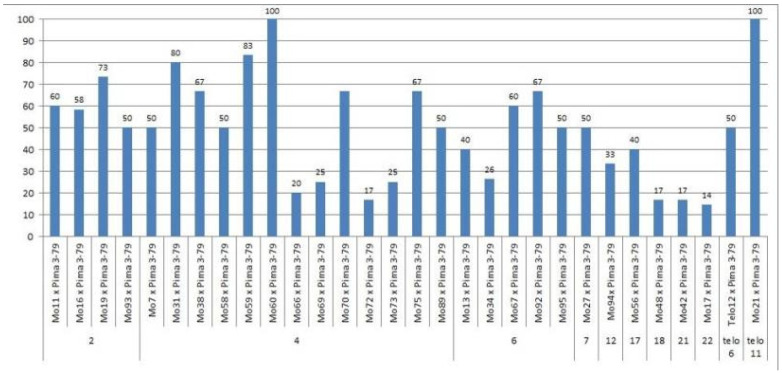
Crossing of aneuploid *G. hirsutum* L. lines with the *G. barbadense* L. line Pima 3-79.

**Figure 3 plants-11-00542-f003:**
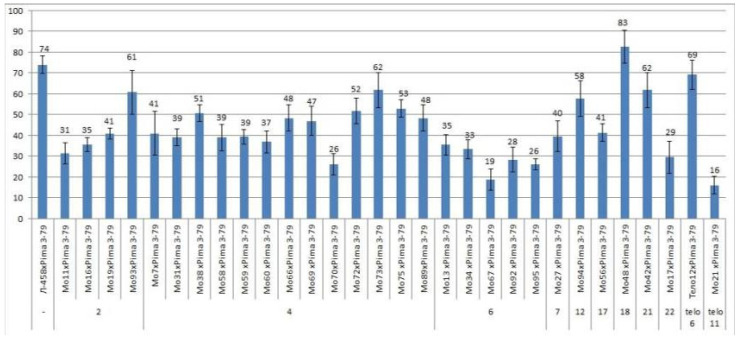
Setting of F_0_ hybrid seeds obtained from crosses of aneuploid *G. hirsutum* L. lines with the *G. barbadense* L. line Pima 3-79.

**Figure 4 plants-11-00542-f004:**
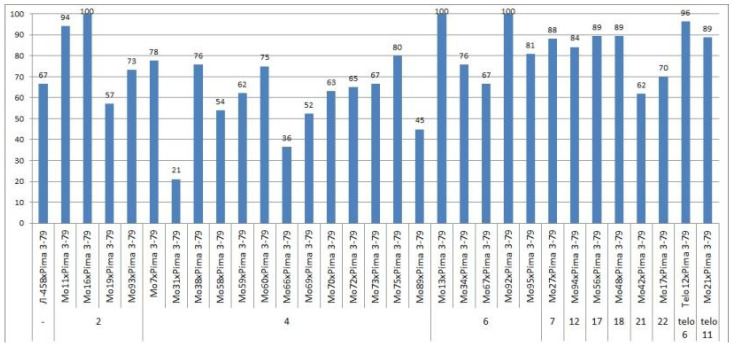
Germination of F_0_ hybrid seeds obtained from crosses of aneuploid *G. hirsutum* L. lines with the *G. barbadense* L. line Pima 3-79.

**Figure 5 plants-11-00542-f005:**
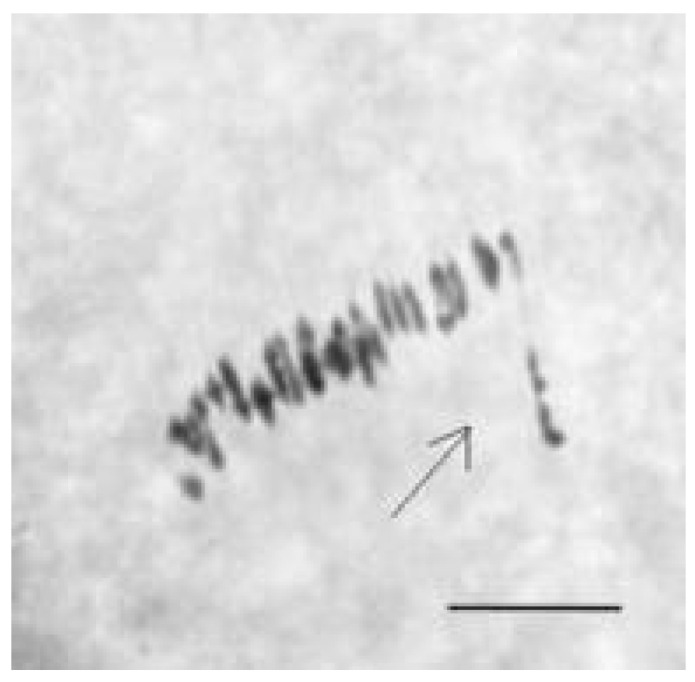
Twenty-five normal bivalents and one heteromorphic bivalent obtained at Metaphase I of F_1_ hybrids from a cross between a monotelodisome line Mo21 × Pima 3-79 (100_1_) for a chromosome **11** arm with the line Pima 3-79. (Heteromorphic bivalent arrowed.) Scale bar = 10 µm.

**Figure 6 plants-11-00542-f006:**
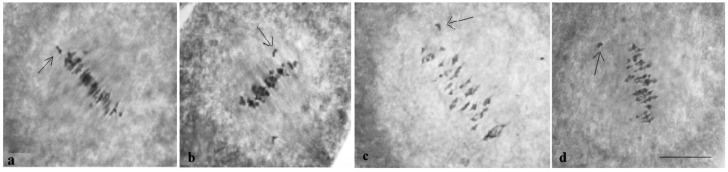
Metaphase I in F_1_ hybrids obtained from crosses of monosomic lines with the line Pima 3-79: (**a**) (Mo16 × Pima 3-79) (25^II^ + 1^I^) (hybrid 98_2_) with monosomy for chromosome **2**; (**b**) (Mo34 × Pima 3-79) (25^II^ + 1^I^) (hybrid 2_8_) and (**c**) (Mo95 × Pima 3-79) (25^II^ + 1^I^) (hybrid 108_3_) with monosomy for chromosome **6**; (**d**) monosomy for chromosome **12** (Mo94 × Pima 3-79) (25^II^ + 1^I^) (8_13_). (Univalents arrowed). Scale bar = 10 µm.

**Figure 7 plants-11-00542-f007:**
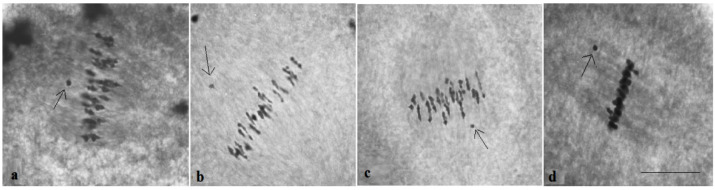
Metaphase I in meiosis of F_1_ hybrids obtained from crosses of monosomic lines with the line Pima 3-79: (**a**) (Mo70 × Pima 3-79) (25^II^ +1^I^) (hybrid 774_6_) and (**b**) (Mo89 × Pima 3-79) (25^II^ + 1^I^) (hybrid 515_7_) with monosomy for chromosome **4**; (**c**) monosomy for chromosome **17** (Mo56 × Pima 3-79) (25^II^ + 1^I^) (4_13_); and (**d**) monosomy for chromosome **18** (Mo48 × Pima 3-79) (25^II^ + 1^I^) (529_16_). (Univalents arrowed.) Scale bar = 100 µm.

**Figure 8 plants-11-00542-f008:**
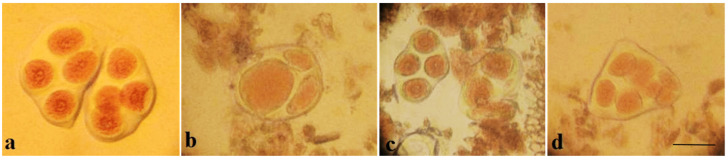
Telophase II in F_1_ hybrids: (**a**) Two tetrads without cytokinesis (Mo16 × Pima 3-79) (98_2_); (**b**) anomalous triad (Mo60 × Pima 3-79) (694_5_); (**c**) dyad with two micronuclei; and (**d**) tetrad with three micronuclei. Scale bar = 40 µm.

**Figure 9 plants-11-00542-f009:**
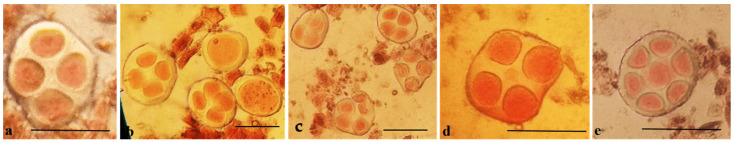
Telophase II in F_1_(Mo67 × Pima 3-79) (308_1_) hybrid: (**a**) normal tetrad; (**b**) two monads and two tetrads; (**c**) anomalous tetrad; (**d**) tetrad with a micronucleus; and (**e**) heptad. Scale bar = 40 µm.

**Figure 10 plants-11-00542-f010:**
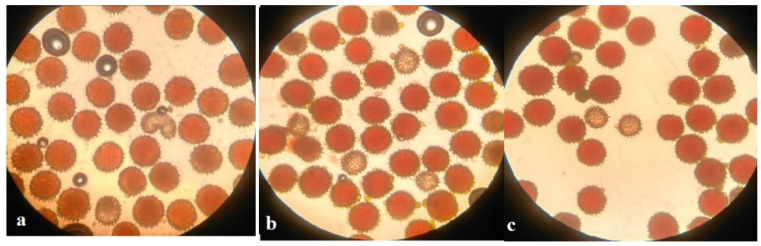
Fertile (coloured) and sterile (uncoloured) pollen grains from a hybrid; (**a**) F_1_ (Mo38 × Pima 3-79) (690_11_) and (**b**,**c**) F_1_(Mo89 × Pima 3-79) (515_7_) for chromosome **4**. Scale bar = 40 µm.

**Figure 11 plants-11-00542-f011:**
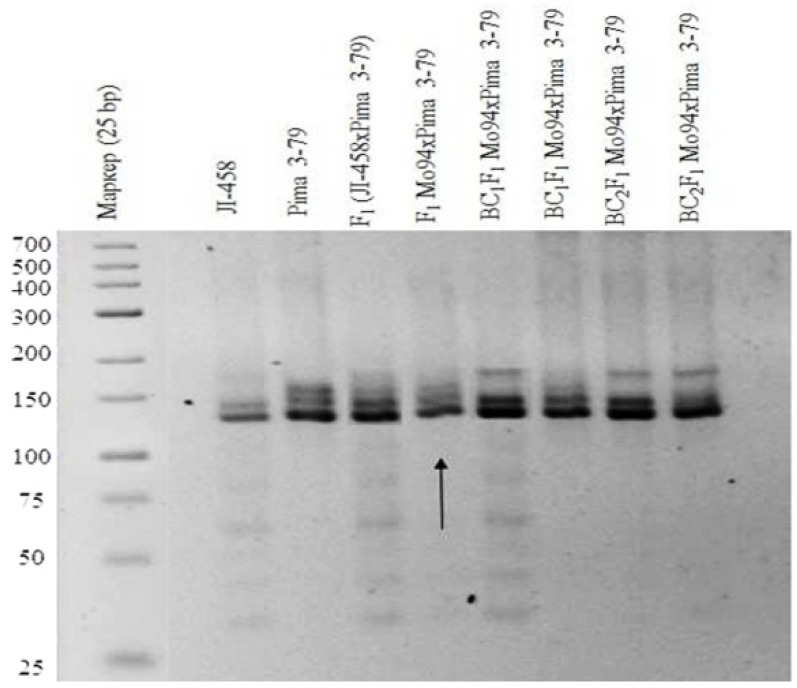
Chromosome identification of monosomic interspecific *G. hirsutum* × *G. barbadense* F_1_ hybrids. SSR primer pair specific for chromosome **12** of the A_t_subgenome: BNL1707.

**Figure 12 plants-11-00542-f012:**
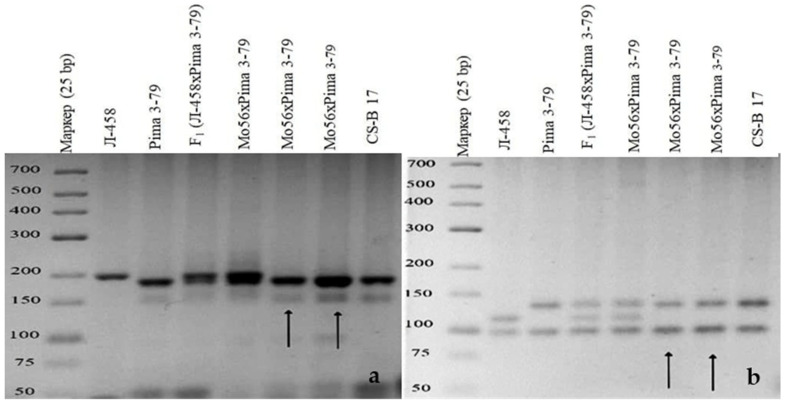
Chromosome identification of monosomic interspecific *G. hirsutum* × *G. barbadense* F_1_ hybrids. SSR primer pair specific for chromosome **17** of the D_t_subgenome: (**a**) TMB0874; (**b**) BNL2496.

**Figure 13 plants-11-00542-f013:**
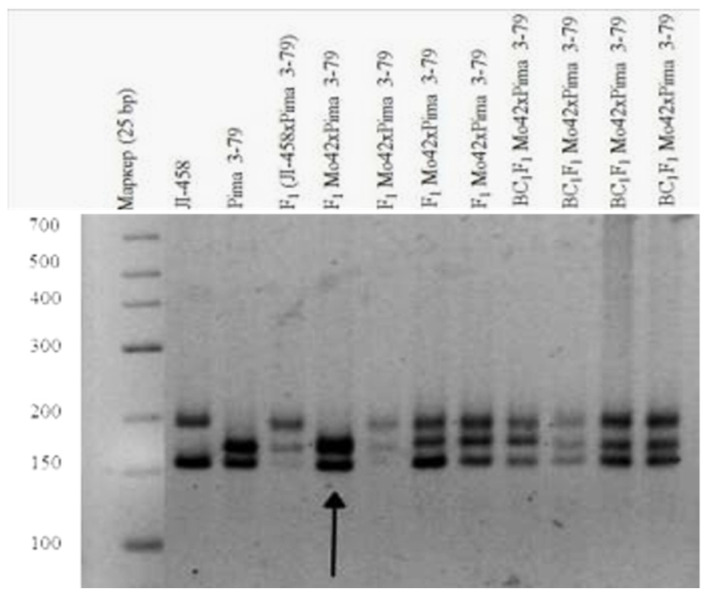
Chromosome identification of monosomic interspecific *G. hirsutum* × *G. barbadense* F_1_ hybrids. SSR primer pair specific for chromosome **21** of the D_t_subgenome: BNL1705.

**Figure 14 plants-11-00542-f014:**
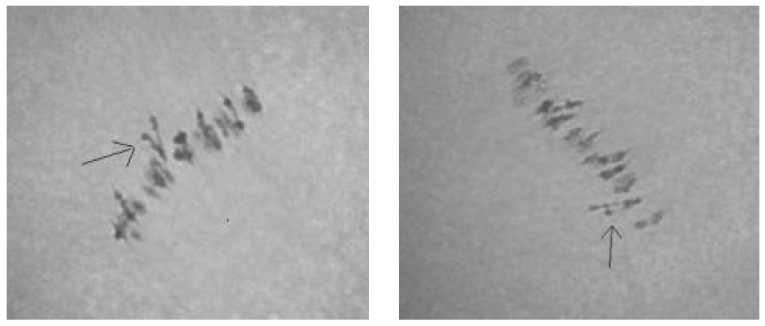
“Critical configurations” of the chromosomes at Metaphase I: 24^II^ + 1^III^ from cotton monosomic F_1_ plants from the cross Mo42 × TT10L-21L. (Arrows point to the trivalents.) Scale bar = 100 µm.

**Figure 15 plants-11-00542-f015:**
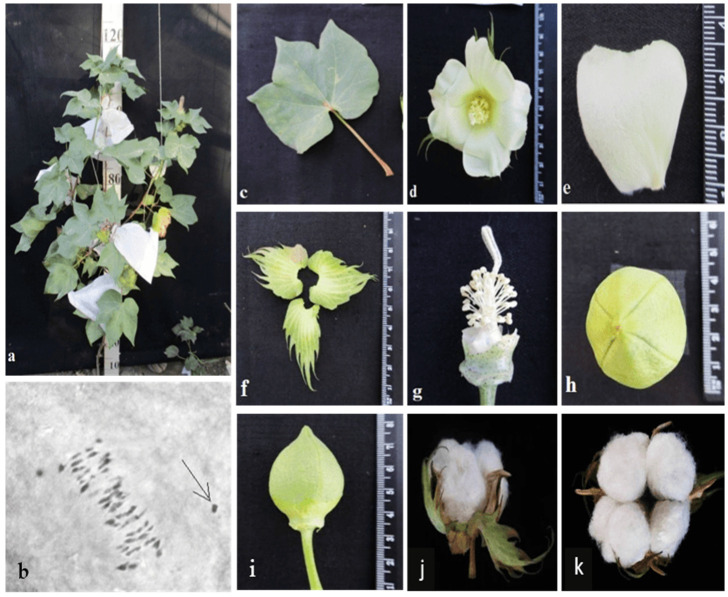
Features of the monosomic cotton lines for chromosome **12**: (**a**) bush, (**b**) configurations of the chromosomes (25^II^ + 1^I^), (**c**) leaf, (**d**) flower, (**e**) petal, (**f**) bracts, (**g**) staminate column, (**h**) and (**i**) green bolls, (**j**) boll with peduncle, and (**k**) open boll.

**Figure 16 plants-11-00542-f016:**
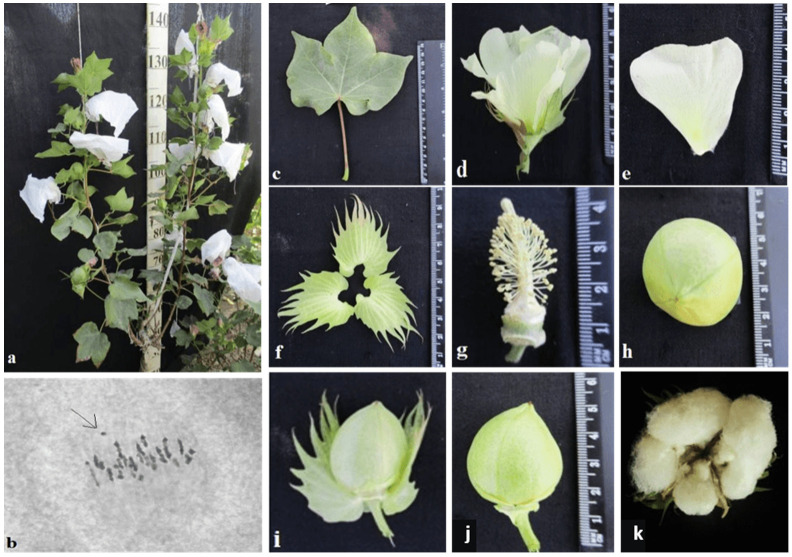
Features of the monosomic cotton lines for chromosome **17**: (**a**) bush, (**b**) configurations of the chromosomes (25^II^ + 1^I^), (**c**) leaf, (**d**) flower, (**e**) petal, (**f**) bracts, (**g**) staminate column, (**h**) and (**i**) green bolls, (**j**) boll with peduncle, and (**k**) open boll.

**Figure 17 plants-11-00542-f017:**
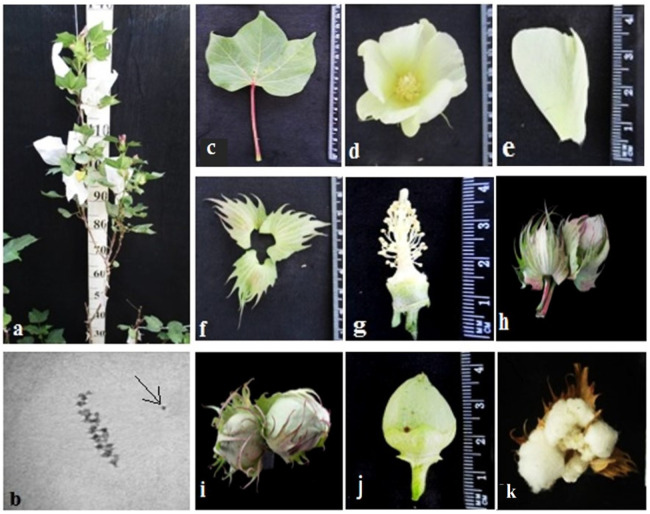
Features of the monosomic cotton line for chromosome **21**: (**a**) bush, (**b**) configurations of the chromosomes (25^II^ +1^I^), (**c**) leaf, (**d**) flower, (**e**) petal, (**f**) bracts, (**g**) staminate column, (**h**) and (**i**) green bolls, (**j**) boll with peduncle, and (**k**) open boll.

**Figure 18 plants-11-00542-f018:**
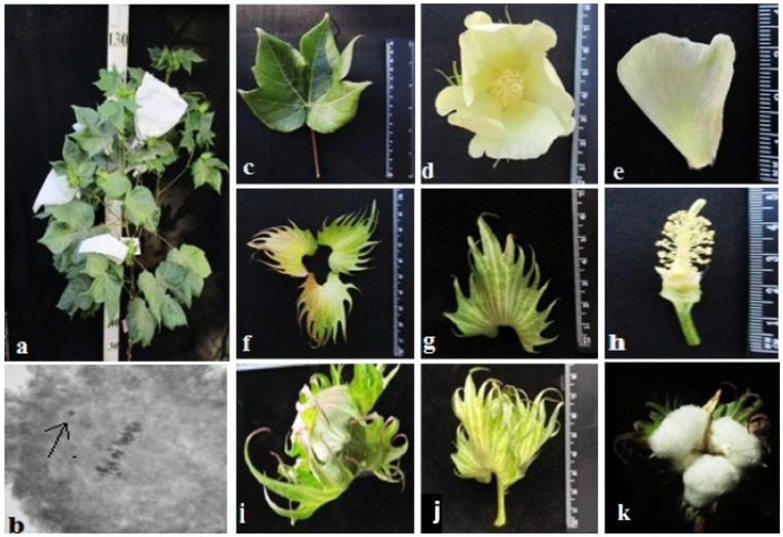
Features of the monosomic cotton line for chromosome **22**: (**a**) bush, (**b**) configurations of the chromosomes (25^II^ +1^I^), (**c**) leaf, (**d**) flower, (**e**) petal, (**f**) bracts, (**g**) bract, (**h**) staminate column, (**i**) green bolls, (**j**) boll with peduncle, and (**k**) open boll.

**Figure 19 plants-11-00542-f019:**
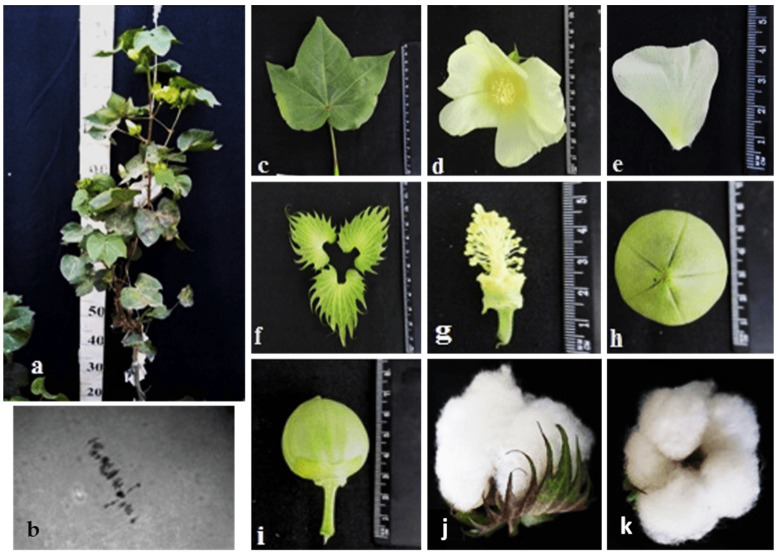
Features of the monotelodisomic cotton lines for chromosome **6**: (**a**) bush, (**b**) configurations of the chromosomes (25^II^ + Ii), (**c**) leaf, (**d**) flower, (**e**) petal, (**f**) bracts, (**g**) staminate column, (**h**) and (**i**) green bolls, (**j**) boll with peduncle, and (**k**) open boll.

**Figure 20 plants-11-00542-f020:**
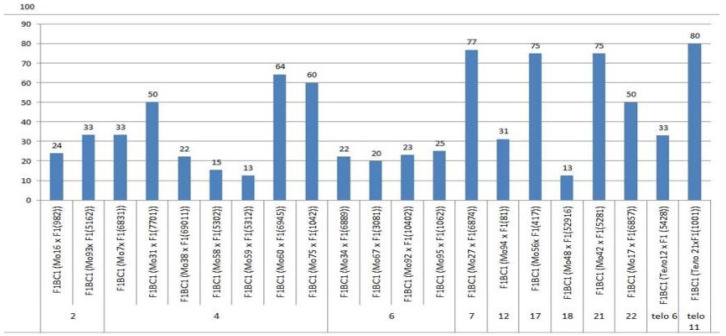
Crossing of aneuploid lines of cotton *G. hirsutum* L. with the interspecific aneuploidy hybrid F_1_ (Mo × Pima 3-79 or Telo × Pima 3-79).

**Figure 21 plants-11-00542-f021:**
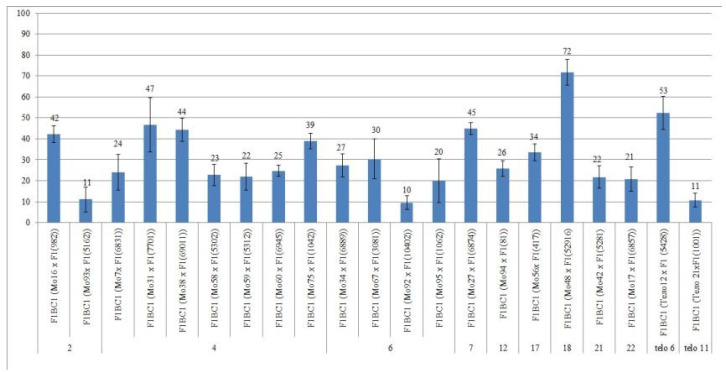
Setting of BC_1_F_1_ hybrid seeds obtained from crosses of aneuploid lines of cotton *G. hirsutum* L. with interspecific aneuploid F_1_ hybrids (Mo × Pima 3-79 or Telo × Pima 3-79).

**Figure 22 plants-11-00542-f022:**
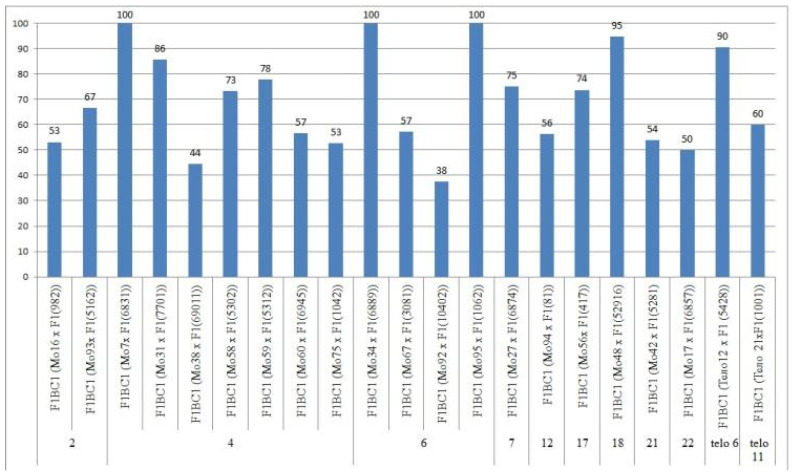
Germination of BC_1_F_1_ hybrid seeds obtained from crosses of aneuploid lines of cotton *G. hirsutum* L. with interspecific aneuploid F_1_ hybrids (Mo × Pima 3-79 or Telo × Pima 3-79).

**Figure 23 plants-11-00542-f023:**
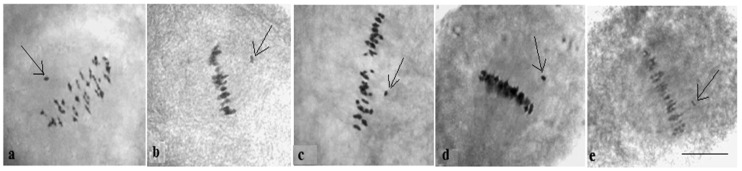
Metaphase I in BC_1_F_1_ hybrids obtained from crosses of monosomic lines with interspecific monosomic F_1_ (Mo × Pima 3-79) hybrids: (**a**) substitution of chromosome **6** (Mo92 × F_1_539_5_) (1040_2_) (25^II^ + 1^I^); (**b**) substitution of chromosome **12** (Mo94 × F_1_8_3_) (299_1_) (25^II^ + 1^I^); (**c**) substitution of chromosome **4** BC_1_F_1_ (Mo58 × F_1_530_3_) (115_1_); (**d**) BC_1_F_1_ (Mo60 × F_1_694_5_) (117_5_); and (**e**) BC_1_F_1_ (Mo75 × F_1_104_2_) (298_3_) (25^II^ + 1^I^). (Univalent arrowed). Scale bar = 10 µm.

**Figure 24 plants-11-00542-f024:**
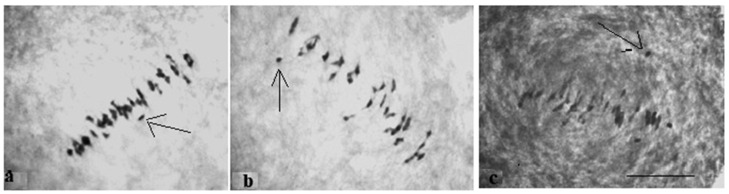
Metaphase I in BC_1_F_1_ hybrids obtained from crosses of monosomic lines with interspecific monosomic F_1_ (Mo × Pima 3-79) hybrids: (**a**) substitution of chromosome **7** (Mo27 × F_1_687_4_) (111_5_) (25^II^ + 1^I^); (**b**) substitution of chromosome **18** (Mo48 × F_1_529_16_) (114_1_) (25^II^ +1^I^); (**c**) substitution of chromosome **22** (Mo17 × F_1_685_7_) (110_1_) (25^II^ + 1^I^). (Univalent arrowed.) Scale bar = 10 µm.

**Figure 25 plants-11-00542-f025:**
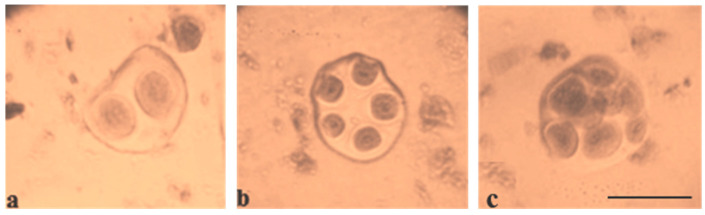
Telophase II in a monosomic BC_1_F_1_ (Mo48 × F_1_529_16_) hybrid (114_1_): (**a**) dyad; (**b**) pentad; (**c**) octad. Scale bar = 40 µm.

**Figure 26 plants-11-00542-f026:**
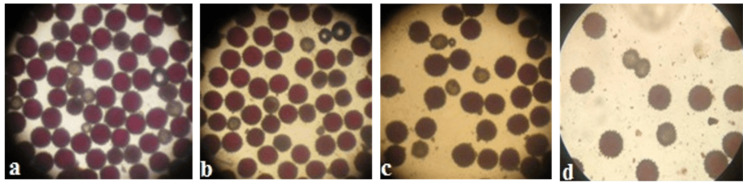
Fertile (coloured) and sterile (uncoloured) pollen grains in a aneuploid hybrid BC_1_F_1_ obtained from a cross between a aneuploid line and a aneuploid hybrid F_1_ (Mo × Pima 3-79 or Telo × Pima 3-79): (**a**,**b**) BC_1_F_1_ (Mo60 × F_1_694_5_) (117_4_); (**c**) BC_1_F_1_ (Mo34 × F_1_688_9_) (293_3_); and (**d**) BC_1_F_1_ (Telo12 × F_1_542_8_) (561_15_).

**Figure 27 plants-11-00542-f027:**
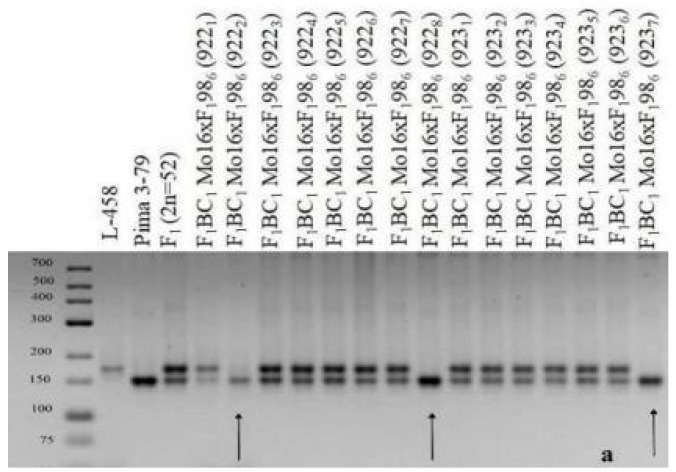
Chromosome identification of monosomic BC_1_F_1_ (Mo16 × F_1_98_6_) hybrids. SSR primer pairs specific for chromosome **2** of the A_t_subgenome JESPR179.

**Figure 28 plants-11-00542-f028:**
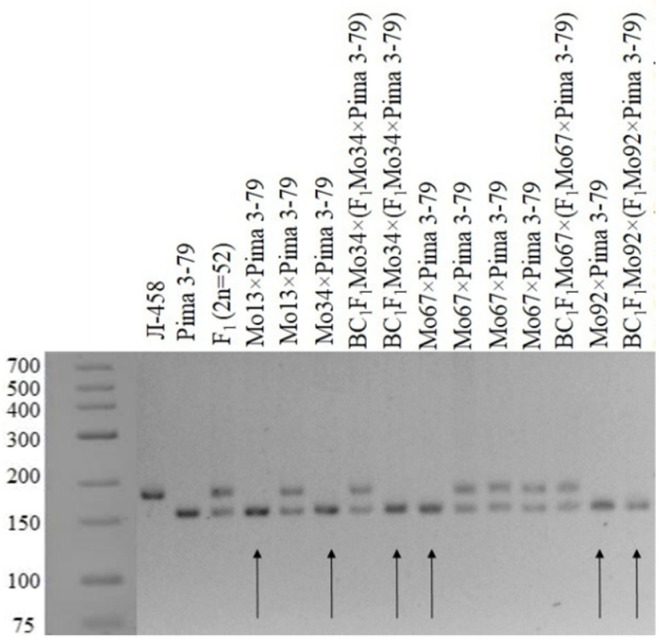
Chromosome identification of monosomic BC_1_F_1_ (Mo34 × F_1_688_9_) hybrids. SSR primer pairs specific for chromosome **6** of the A_t_subgenome Gh082.

**Figure 29 plants-11-00542-f029:**
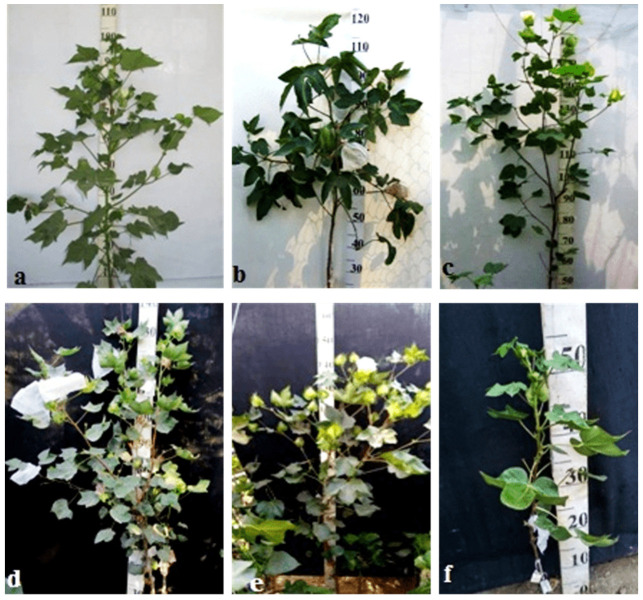
Plants of the parental lines L-458 (**a**) and Pima 3-79 (**b**) disomic F_1_ hybrid L-458 × Pima 3-79 (**c**) (Mo × Pima 3-79) and F_1_BC_1_ obtained from crosses of the recurrent parent; (**d**) monosomic line for chromosome **2**, Mo16; (**e**) F_1_(Mo16 × Pima 3-79) (98_2_); and (**f**) substitution of chromosome **2** BC_1_F_1_(Mo16 × F_1_98_2_) (923_7_).

**Figure 30 plants-11-00542-f030:**
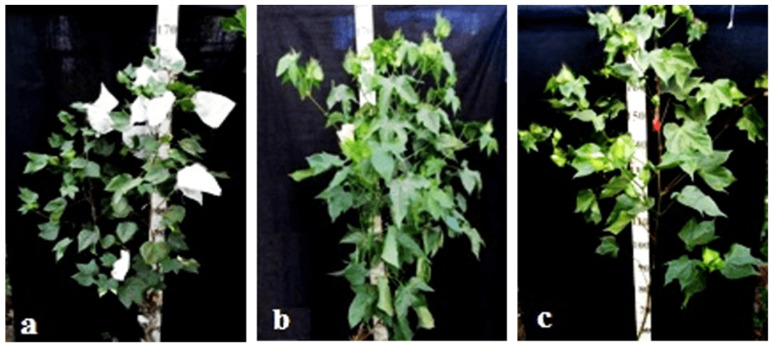
Plants of monosomic line *G. hirsutum* for chromosome **4**, Mo59 (**a**); F_1_Mo59 × Pima 3-79 (531_2_) (**b**); and BC_1_F_1_ (Mo59 × F_1_531_2_) (1041_4_) with substitution for chromosome **4**
*G. barbadense* (**c**).

**Figure 31 plants-11-00542-f031:**
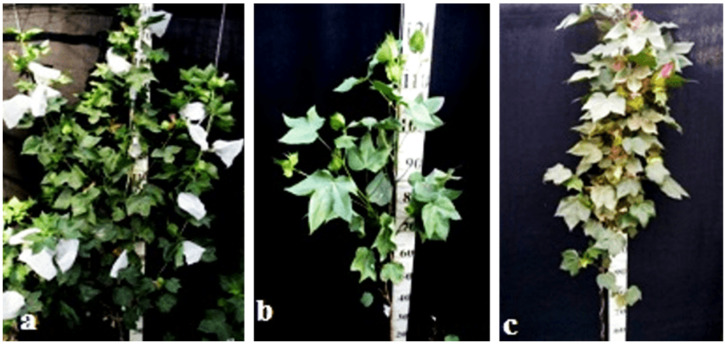
Plants of monosomic line *G. hirsutum* for chromosome **6**, Mo34 (**a**); F_1_Mo34 × Pima 3-79 (688_9_) (**b**); and BC_1_F_1_ (Mo34 × F_1_688_9_) (293_3_) with substitution for chromosome **6**
*G. barbadense* (**c**).

**Figure 32 plants-11-00542-f032:**
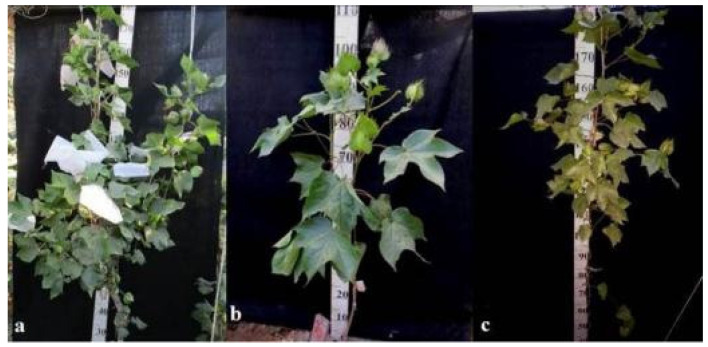
Plants of monosomic line *G. hirsutum* for chromosome **7**, Mo27 (**a**); F_1_Mo27 × Pima 3-79 (687_4_) (**b**); and BC_1_F_1_ (Mo27 × F_1_687_4_) (111_2_) with substitution for chromosome **7**
*G. barbadense* (**c**).

**Figure 33 plants-11-00542-f033:**
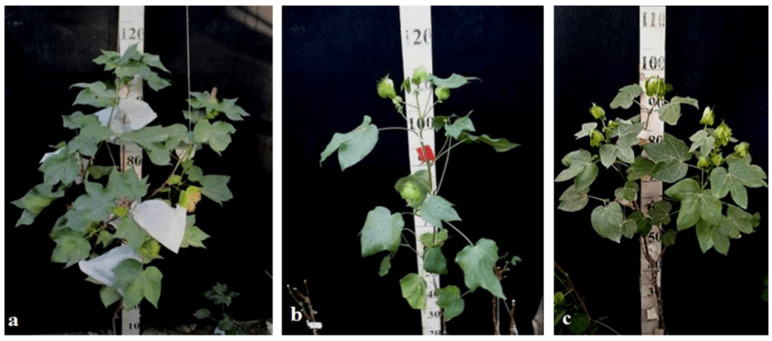
Plants of monosomic line *G. hirsutum* for chromosome **12**, Mo94 (**a**); F_1_Mo94 × Pima 3-79 (8_1_) (**b**); and BC_1_F_1_ (Mo94 × F_1_8_1_) (299_1_) with substitution for chromosome **12**
*G. barbadense* (**c**).

**Figure 34 plants-11-00542-f034:**
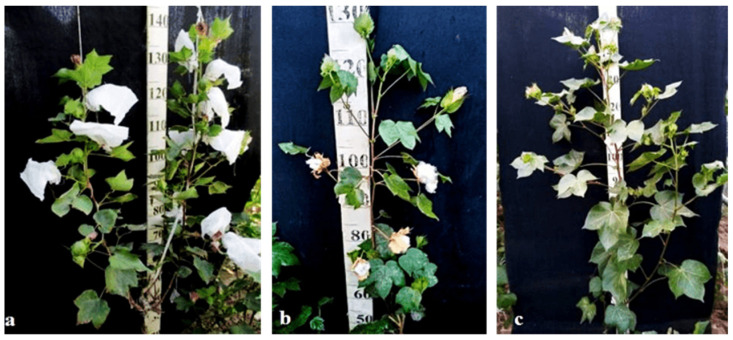
Plants of monosomic line *G. hirsutum* for chromosome **17**, Mo56 (**a**); F_1_Mo56 × Pima 3-79 (4_17_) (**b**); and BC_1_F_1_ (Mo56 × F_1_4_17_) (513_8_) with substitution for chromosome **17**
*G. barbadense* (**c**).

**Figure 35 plants-11-00542-f035:**
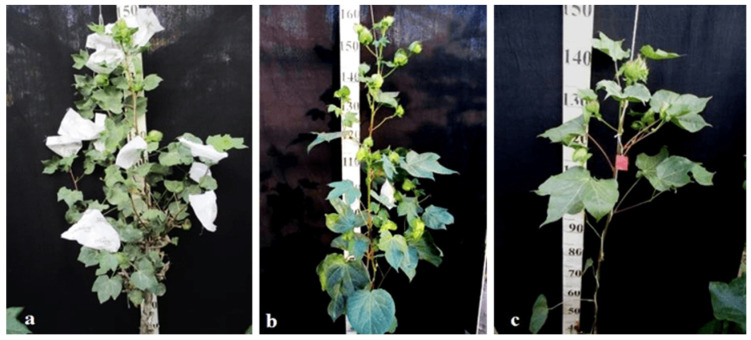
Plants of monosomic line *G. hirsutum* for chromosome **18**, Mo48 (**a**); F_1_Mo48 × Pima 3-79 (529_16_) (**b**); and BC_1_F_1_ (Mo48 × F_1_529_16_) (114_20_) with substitution for chromosome **18**
*G. barbadense* (**c**).

**Figure 36 plants-11-00542-f036:**
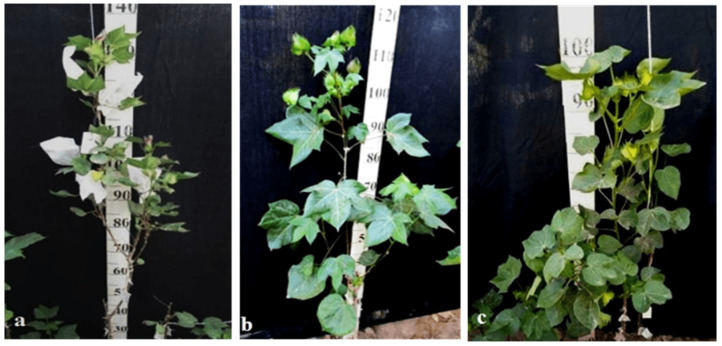
Plants of monosomic line *G. hirsutum* for chromosome **21**, Mo42 (**a**); F_1_ Mo42 × Pima 3-79 (528_1_) (**b**); and BC_1_F_1_ (Mo42 × F_1_528_1_) (782_1_) with substitution for chromosome **21**
*G. barbadense* (**c**).

**Figure 37 plants-11-00542-f037:**
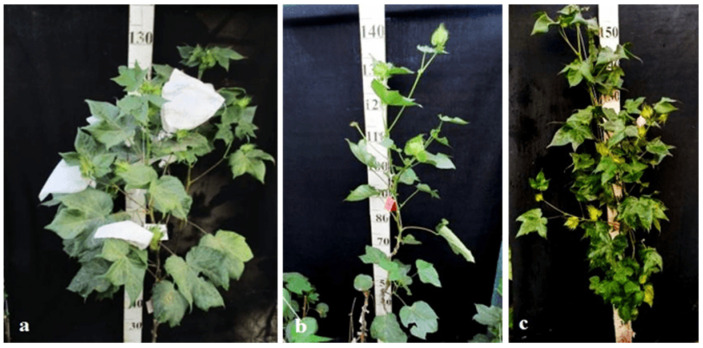
Plants of monosomic line *G. hirsutum* for chromosome **22**, Mo17 (**a**); F_1_ Mo17 × Pima 3-79 (685_7_) (**b**); and BC_1_F_1_ (Mo17 × F_1_685_7_) (110_1_) with substitution for chromosome **22**
*G. barbadense* (**c**).

**Figure 38 plants-11-00542-f038:**
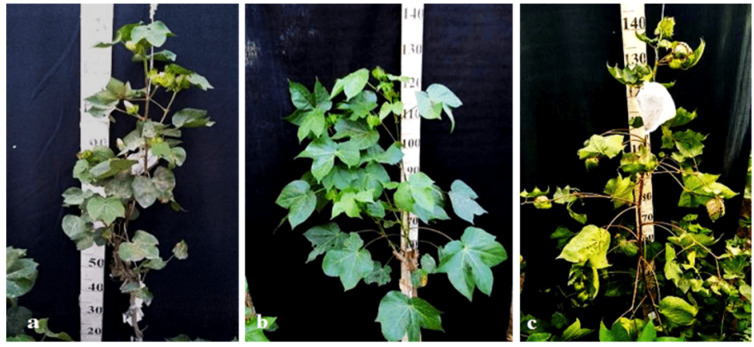
Plants of the parental line and monotelodisome F_1_ and F_1_BC_1_ cotton hybrids obtained from crosses of the recurrent parent with monotelodisome F_1_ hybrids (Telo12 × Pima 3-79): (**a**) monotelodisome line Telo12, chromosome **6**; (**b**) F_1_ Telo12 × Pima 3-79 (542_16_); and (**c**) BC_1_F_1_ (Telo12 × F_1_542_16_) (561_15_) with substitution of an arm of chromosome **6**.

**Figure 39 plants-11-00542-f039:**
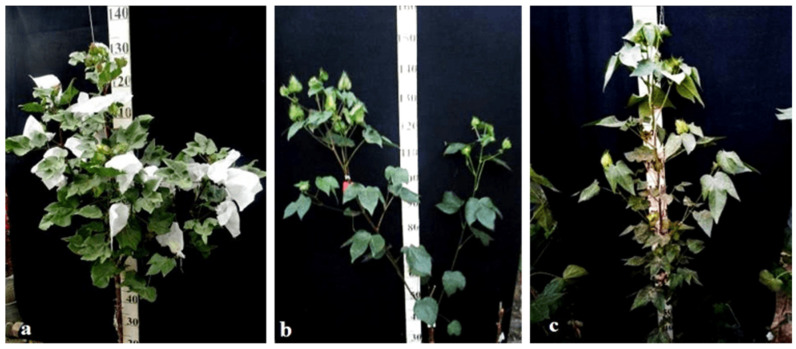
Plants of the parental line and monotelodisomic F_1_ and F_1_BC_1_ hybrids obtained from crosses of the recurrent parent with monotelodisome F_1_ hybrids (Telo × Pima 3-79): (**a**) monotelodisome line Mo21, chromosome **11**; (**b**) F_1_Mo21 × Pima 3-79 (102_1_); and (**c**) BC_1_F_1_(Mo21 × F_1_102_1_) (292_1_) with substitution of an arm of chromosome **11**.

**Table 1 plants-11-00542-t001:** Microsatellite markers used in the analysis of monosomic interspecific hybrids of cotton.

Chromosome	SSR Marker	Collection	Size of the PCR Products for each SSR Marker, n. b. p. *		Literature Source
			** *G. hirsutum* ** **(L-458)**	** *G. barbadense* ** **(Pima 3-79)**	
12	BNL1227	BNL	157, 186	171	[36]
	BNL3835	BNL	193	185	[36]
	BNL1707	BNL	144, 154	144, 153, 157	[36]
	BNL3594	BNL	182, 232	184, 200	[36]
	BNL3886	BNL	229	200, 231	[36]
	BNL3261	BNL	203	195	[36]
17	BNL1606	BNL	191	175, 187	[36]
	BNL2496	BNL	96, 112	96, 129	[36]
	BNL3371	BNL	167, 194	167, 189	[36]
	BNL3955	BNL	165	156, 189	[36]
	BNL2471	BNL	195	191	[36]
	TMB0874	TMB	198	187	[37]
	TMB2018	TMB	228, 231	228, 256	[37]
21	BNL1705	BNL	192	168	[36]
22	BNL673	BNL	192	190	[36]
telo 6	BNL2884	BNL	192	190	[36]
	TMB0154	TMB	258	246	[37]

* n. b. p.—nucleotide base pairs.

**Table 2 plants-11-00542-t002:** Origin and some features of monosomic *G. hirsutum* L. lines.

Monosomic Line	Origin	Year Obtained	Chromosome		Morphological Characteristics
			Size	Identity	
Mo94	Pollen irradiation at a dose of 20 Gy in M_3_	2003	Large	**12** A_t_ subgenome	Shortened sympodial branches, leaves with slightly curved edges, small bolls, short pedicels (1–1.5 cm), small bracts (3.5–5 cm), a small number of stamens (from 55 to 75), and the absence of external nectaries, with internal nectaries rarely present.
Mo56	Irradiation by thermal neutrons at a dose of 35 Gy in M_1_	1995	Medium- small	**17** D_t_ subgenome	Compact bush, small leaves, short petioles, small ovoid bolls, abundant budding but weak seed setting, small flowers, short stalks (up to 1 cm), small bracts (3.2–4.0 cm), a small number of teeth in the bracts (from 9 to 15 pcs.), the absence of internal nectaries and the rare presence of external nectaries, and a small number of stamens (from 60 to 70).
Mo42	Pollen irradiation at a dose of 20 Gy in M_1_	1994	Medium-small	**21** D_t_ subgenome	Compact bush, small three-lobed leaves, shortened sympodial branches, small ovoid bolls, sometimes of the cluster type, abundant budding, small flowers, short stalks (up to 1–1.4 cm), small bracts (2.5–4.5 cm), a small number of teeth in the bracts (from 10 to 14 pcs.), average number of stamens (104 pcs.), and absence of external and internal nectaries.
Mo17	Pollen irradiation at a dose of 25 Gy in M_1_	1991	Medium-small	**22** D_t_ subgenome	Lush bush, elongated leaf blades, elongated bracts and pedicels, elongated ribbed bolls, medium stalks (up to 1.6 cm), small bracts (3.5–4.5 cm), a small protrusion of the stigma (0.5 cm), and small bolls.
Telo12	Pollen irradiation at a dose of 20 Gy to progeny Mo21 in M_2_	2000	Large	Telo **6** A_t_ subgenome	Compact bush, abundant budding, spherical bolls, small peduncles (1 cm) and bracts (3.5–4.5 cm), a small number of teeth on the bracts (9 to 15 pcs.), stamens (60 to 70), and a small protrusion of the stigma (0.6–1 cm).

**Table 3 plants-11-00542-t003:** Analysis of microsatellite markers (SSRs) across cotton families.

Chromosome	Hybrid	Family	Plant Number	Presence of Markers for *G. barbadense* L.
2	BC_1_F_1_(Mo16 × F_1_98_6_)	Family 1	922_1_	
			922_2_	BNL834, TMB0471, JESPR179
			922_3_	
			922_4_	
			922_5_	
			922_6_	
			922_7_	
			922_8_	BNL834, TMB0471, JESPR179
		Family 2	923_1_	
			923_2_	
			923_3_	
			923_4_	
			923_5_	
			923_6_	
			923_7_	BNL834,TMB0471, JESPR179
4	BC_1_F_1_(Mo31 × F_1_770_1_)	Family 1	924_1_	TMB0809, Gh117, BNL2572, Gh107
			924_2_	
			924_3_	
			924_4_	
			924_5_	
			924_6_	
	BC_1_F_1_(Mo38 × F_1_690_11_)	Family 1	925_1_	
			925_2_	
			925_3_	
			925_4_	BNL2572, Gh107, TMB0809, Gh117
			925_5_	
			925_6_	
			925_7_	BNL2572, Gh107, TMB0809, Gh117
			925_8_	
			925_9_	
			925_10_	
			925_11_	BNL2572, TMB0809, Gh117

**Table 4 plants-11-00542-t004:** Analysis of microsatellite markers (SSRs) in cotton interspecific BC_1_F_1_ hybrids.

Chromosome	Hybrid	Hybrid Plant Number	Presence of Markers for *G. barbadense*
4	BC_1_F_1_Mo60 × F_1_694_5_	117_5_	Gh107, Gh117, TMB0809,CIR249, JESPR234
	BC_1_F_1_Mo75 × F_1_104_2_	298_2_	BNL2572, Gh107, Gh117, TMB0809, CIR249, JESPR234
		298_3_	BNL2572, Gh107, Gh117, TMB0809, CIR249, JESPR234
6	BC_1_F_1_Mo34 × F_1_688_9_	293_3_	BNL1440, BNL2884, BNL3359, BNL3650, TMB0154, TMB0853, TMB1277, TMB1538, Gh039, Gh082
	BC_1_F_1_Mo92 × F_1_539_5_	1040_2_	BNL1440, BNL2884, BNL3359, BNL3650, TMB0154, TMB0853, TMB1277, TMB1538, Gh039, Gh082
12	BC_1_F_1_Mo94 × F_1_8_1_	299_1_	BNL1227, BNL3835
	BC_1_F_1_Mo94 × F_1_8_1_	299_2_	BNL3594
22	BC_1_F_1_Mo17 × F_1_685_7_	110_3_	BNL673

**Table 5 plants-11-00542-t005:** Microsatellite markers used in the analysis of cotton monosomic interspecific BC_1_F_1_ hybrids.

Chromosome	SSR Marker	Collection	Size of the PCR Products for each SSR Marker, n. b. p. *		Literature Source
			***G. hirsutum* L.** **(L-458)**	***G. barbadense* L.** **(Pima 3-79)**	
2	BNL834	BNL	120	-	[36]
	BNL3971	BNL	140	116	[36]
	JESPR179	JESPR	174	152	[40]
	TMB0471	TMB	213	173	Detected using GelAnalyzer 19.1 software
4	BNL2572	BNL	248	234	[36]
	Gh107	Gh	380	280	[38]
	Gh117	Gh	260	240	[38]
	TMB0809	TMB	208	146	Detected using GelAnalyzer 19.1 software
6	BNL1440	BNL	256	266	[36]
	Gh039	Gh	125	120	[38]
	Gh082	Gh	175	155	[38]
	TMB0154	TMB	258	246	[37]
	TMB0853	TMB	249	237	[37]
	TMB1277	TMB	251	263	[37]
	TMB1538	TMB	208	196	[37]
12	BNL1227	BNL	157, 186	171	[36]
	BNL3835	BNL	193	185	[36]
	BNL1707	BNL	144, 154	144, 153, 157	[36]
	BNL3594	BNL	182, 232	184, 200	[36]
	BNL3886	BNL	229	200, 231	[36]
	BNL3261	BNL	203	195	[36]
17	BNL1606	BNL	191	175, 187	[36]
	BNL2496	BNL	96, 112	96, 129	[36]
	BNL3371	BNL	167, 194	167, 189	[36]
	BNL3955	BNL	165	156, 189	[36]
	BNL2471	BNL	195	191	[36]
	TMB0874	TMB	198	187	[37]
	TMB2018	TMB	228, 231	228, 256	[37]
21	BNL1705	BNL	192	168	[36]
22	BNL673	BNL	192	190	[36]

## Data Availability

Data are contained within the article or the Appendix A.

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
