# Peer review of "Features of Chromosome Introgression from Gossypium barbadense L. into G. hirsutum L. during the Development of Alien Substitution Lines"

_plants, 2022, doi:10.3390/plants11040542_

Round 1
Reviewer 1 Report
In general, this manuscript addresses features of chromosome introgression from Gossypium barbadense L. into Gossypium hirsutum L. during the development of alien substitution lines. I have the following comments that hopefully will be useful for improving this manuscript.
- Although the author made a large number of experiments to analyze features of chromosome introgression from Gossypium barbadense L. into Gossypium hirsutum L. during the development of alien substitution lines from phenotype, cytology, molecular markers and other levels, the structure of the article was chaotic and the text description was not clear. For example, in the introduction, many research contents are irrelevant to this study, so we suggest deleting them. The result part of the article is too long, it is suggested to be appropriately deleted.
- Many figures provided in this manuscript were improperly edited by PS or other software. Obviously, the marker lines in the figure 11,12,13,30,31b,32,33,34 were cut from other pictures
- Bars in many figures were missing, such as 16, 17, 18,1 9, 20.21.
- The reference format is not uniform, some cited as numbers and some as names.
- The format of the reference list in the reference part was not uniform too.
- The words on the picture are not clear. Arrows in the figures overlapped with the text. Numbers in the figures 2, 3, 23 were not clear.
- Picture layout is not beautiful, such as 17, 18, 19, 20, 21. These figures can be combined into one.
- Many typos in the manuscript, such as metaphaseI (P8L221), detecte (p13l327). Symbol “x” showing hybridization between two materials should be “×”.
Author Response
Reply to reviewer 1:
1) We agreed with the opinion of the reviewer and removed from the introduction the results of those studies that are not directly related to the topic of the article.
2) We have edited the recommended figures in the PS program.
3) We have edited the recommended figures with the results of electrophoresis.
4) We tried to make all the stripes clearer.
5) We made all references uniform and cited all authors as numbers.
6) We made changes and made the list of references uniform.
7) We have redesigned all the diagrams and made them easy to view.
8) We redid all the figures of the new lines and made them into one picture.
9) We corrected all typos in the manuscript and made changes to the spelling of the symbol.

Reviewer 2 Report
Chromosome substitution line population is one of the important tools to transfer and analyze agronomical valuable traits from relative species to cultivated crops. Partial aneuploidy in cotton have been obtained, which provide the possibilities for developing substitution line population at the chromosome level. But cotton cytogenetic research lag far behind wheat duo to its numerous chromosomes with similar sizes and the lacking of technique for chromosome discrimination. The authors analyzed the features of chromosome introgression during the development of G. hirsutum-G. barbadense alien substitution lines using molecular markers and chromosome paring behavior at meiosis metaphase I stage. They performed huge amount of cytogenetic work and obtained a great amount of data. They successfully developed several new chromosome substitution lines of G. hirsutum-G. barbadense, which would serve as sources of introgression of new traits and properties from G. barbadense to G. hirsutum one by one at the chromosomal level. I strongly recommend the article for publication after several minor modifications.
- Section of Introduction is too long, it should be simplified from 167 lines to within 100 lines.
- Section of Discussion is too long about 12 pages, it should be within 5 pages.
- Lines 49-50, the description of “Genome sizes vary from large to small in the following order: A>F>B>E>C>G>K>D [5]” seems incorrect. K genome size is the largest (about 2.5 Gb), while A genome size is about 1.8 Gb. Please refer to new data and check it.
- As we know, A chromosomes are larger than D, theoretically, losing A chromosome should has greater effects on crossing rate or pollen vitality than losing D chromosome. Interestingly, the author found three lines with deficiencies in chromosomes from the Dt subgenome were characterized as having the largest decrease in crossing rate, for instance, three lines (Mo48, Mo42 and Mo17) with deficiencies in chromosomes 18, 21 and 22 from the Dt subgenome. It is common phenomenon or an exception? Please explain the phenomenon.
- How to calculate the meiotic index, please add the formula.
- In Table 2, Chromosome Identity, D17, D21 and D22, the terms are not proper.
- Many figures should be rearranged into a single figure for comparison of different chromosome effects according their morphological traits, chromosome behaviors, and gel images.
- Please go through the whole manuscript and correct spelling errors because many two or three words are combined into one, which should be split by a half space.
Author Response
Reply to reviewer 2.
1. We have significantly shortened the introduction section, as requested by the reviewer.
2. We have significantly reduced the discussion section, as requested by the reviewer.
3. We agree with the reviewer's opinion about the inaccuracy in the sizes of various genomes in cotton, given on lines 49-50, since in a later work (Wendel et al., 2015), the sizes of genomes in the genus Gossypium are arranged in the following order: K genome (2570 Mb)> C(1980 Mb)> G (1785 Mb)> A (1700 Mb)> E (1560 Mb)> B> (1350 Mb) F (1300 Mb)> D (885 Mb). However, due to the reviewer's request to shorten the introduction section, we considered it possible to remove this sentence from the text of the article.
4. We think that the decrease in crossability in three monosomic lines (Mo48, Mo42 and Mo17) with deficiencies of chromosomes 18, 21 and 22 of the D-subgenome is an exceptional feature of these specific chromosomes of the D-subgenome, since our collection contains the identified monosomic line Mo56 with a lack of chromosome 17 of the D-subgenome with a higher crossability (40.00%). Indirect confirmation of this statement is the discovery in our collection of two other monosomic lines with deficiencies of chromosomes 17 and 25 of the D-subgenome, which are characterized by crossing rates of 45.45 and 57.14%, respectively (unpublished data). Identification of univalent chromosomes in other lines of our collection, as well as clarification of their crossing with donor lines in the future, will allow a more thorough and specific answer to your question. However, it seems to us that the chromosomes of the D-subgenome of cotton may reveal many new surprises in the future, since the localization of most QTL loci in the D-subgenome, the species of which do not have a spinning fiber at all, is one of the unexpected surprises of this genome.
5. The meiotic index is calculated as the number of normal tetrads from the total number of different types of sporads studied, expressed as a percentage. We have included the corresponding formula for calculating the meiotic index in the Material and Methods section.
6. Since the brief reduction in the numbers of univalent chromosomes did not satisfy the reviewer, we corrected these designations and wrote in Table 2 the full chromosome designations as 17 Dt subgenome, 21 Dt subgenome, and 22 Dt subgenome.
7. We agree with the opinion of the reviewer and have converted the figures in the article into one large figure.
8. We reviewed the manuscript of the article, corrected spelling errors and separated all the words in the article.

Reviewer 3 Report
The text constantly contains mistakenly merged words,
Figures 5, 6, 7, 8, 9, 10, 25, 27, 28 and 29 lack the scale size
Author Response
Reply to reviewer 3.
- We made corrections to the text and separated all erroneously merged words.
- We have indicated in figures 5, 6, 7, 8, 9, 10, 25, 27, 28 and 29 the corresponding scale.